# Some aspects of the deep abyssal overflow between the middle and southern basins of the Caspian Sea

**Javad Babagoli Matikolaei[1], AbbasaliAliakbariBidokhti[2], and Maryam Shiea[3]**

[1]Graduate in Physical Oceanography, Institute of Geophysics, University of Tehran, Tehran, Iran.
[2]Institute of Geophysics, University of Tehran, Tehran, Iran.
[3]Faculty of Marine Science and Technology, Science and Research Branch, Islamic Azad University, Tehran, Iran.

*Correspondence to*: Abbasali AliakbariBidokhti(bidokhti@ut.ac.ir)

**Abstract.**This study investigates the deep gravity current between the middle and southern Caspian Sea basins, caused by density difference of deep waters. Oceanographic data, numerical model, and dynamic models are used to consider the structure

of this Caspian Sea abyssal overflow. The CTD data are obtained from UNESCO, and the three-dimensional ocean model COHERENS results are used to study the abyssal currents in the southern basin of the Caspian Sea.

The deep overflow is driven by the density difference mainly due to the temperature difference between the middle and southern basins, especially in winter. Due to Cold weather in the northern basin, water sinks in high latitudes and after filling the middle basin it overflows into the southern basin. As the current passes through the Absheron Strait (or sill), we use an

analytic model for the overflow gravity current to estimate the changes of vorticity and potential vorticity of the flow over the Absheron sill, we use the method of Falcini and Salusti (2015) in which the effects of entrainment and friction are considered. Because of the importance of the overflow in deep water ventilation, a simple dynamical model of the boundary currents based on the shape of the Strait is used to estimate typical mass transport and flushing time which is found to be about 15 to 20 years for the southern basin of the Caspian Sea. This time scale is important for this ecosystem and impacts of pollution due to oil

exploration. Separate from this, by reviewing the oil and gas wells of the Caspian Sea, the results show that the deep flow moves exactly the path of this wells. Thus, the deep flow can be one of important reason of oil pollution in the depth of southern Caspian Sea.

**Keywords:** Overflow, dynamical model, trapped baroclinic bottom current, Caspian Sea abyssal flow.

# 1. Introduction

Baroclinic flows play important roles in the ocean and sea circulations, especially in deep waters of the ocean. Because these currents are important in deep water ventilations of the oceans, they have an integral role in thermohaline circulation. A driving mechanism for the circulation is cooling of surface waters at high latitudes and consequent formations of deep waters by sinking the cooled salty water masses (Fogelqvist et al., 2003).

Cooling in polar seas (Dickson et al., 1990) and evaporation in marginal seas (Baringer and Price, 1997) cause dense waters that sink to form deep water masses. For example, dense water from the deep convective regions of the North Atlantic produces a signature of the thermohaline overturning circulation that can be seen as far away as the Pacific and Indian oceans (Girton et al., 2003). In the global sense, bottom-trapped currents play an integral role in thermohaline circulation and are a vehicle for the transport of heat, salt, oxygen,and nutrients over great distances and depths. Mixing and exchange processes between the along-slope currentsat the continental shelves and deep ocean water can also affect the thermohaline circulations. Huthnance (1995) has reviewed the processes involved in such near-shelf circulations. He has pointed out such flows around world that may lead toformation of mesoscale eddies as they become unstable while moving along the sloped boundary. The ability of abyssal flows to transport and deposit sedimentsis also of geological interest (Smith, 1975).

As thermohaline circulation causes ventilation of deep ocean water, it is important not only in open seas and ocean but also in semi-closed and closed basins ventilations, e.g. the Caspian Sea. Study of thermohaline dynamics and circulation has also been of interest to other scientists such as climate researchers. The dynamics of such dense currents onslopes have been modelled in the past both theoretically and experimentally starting with Ellison and Turner (1959) and Britter and Linden (1980), and a review on gravity currents can be found in Griffiths (1986).

The Caspian Sea, the world's largest inland enclosed water body, consists of three basins namely northern(shallow, mean depth of about 10 m and covering 80000 km$^2$), the middle (rather deep, with mean depth of about 200 m, maximum depth 788 m and covering 138000 km$^2$) and the southern (deep, with a mean depth of 350 m, maximum depth 1025 m and covering 164840 km$^2$) and is located between 36.5° Nand 47.2° N, and 46.5° E and 54.1° E (Aubrey et al., 1994; Aubrey, 1994). The depth varies greatly over this sea (Ismailova, 2004, Figure 1). The northern basin, after a sudden depth transition at the shelf edge, reaches the middle one. The middle and southern basins are divided by the Absheron sill or Strait (with a maximum depth of 180m).The western slopes of the two deeper basins are fairly steep compared to the eastern slope (Gunduz and Özsoy, 2014). Peeters et al (2000) have also estimated the ages of waters of the Caspian Sea basins while considering the exchange between the middle and southern basins, based on chemical tracers, and found typical ages of about 20 to 25 years depending on the exchange rates. This exchange rate seems to vary year by year and is dominated by atmospheric forcing.

The Caspian Sea is enclosed with tides being fairly weak and circulation in this sea is mainly due to wind and buoyancy, although some wave-driven flows also occur in coastal regions (Bondarenko, 1993; Ghaffari and Chegini, 2010; Ghaffari et al., 2013; Ibrayev et al., 2010; Terziev et al., 1992). The seasonal circulation based on a coupled sea hydrodynamics, air-seainteraction and sea ice thermodynamics model of the Caspian Sea was investigated by Ibrayev et al.(2010) andGunduz and

Özsoy(2014).The effect of fresh water inflow to the Caspian Sea on seasonal variations of salinity and circulation (or flow) pattern of the Caspian Sea surface has also been studied using HYCOM model (Kara et al, 2010). These studies indicate that in north-eastern parts of the middle basin of this sea there are signs of sinking water in cold season. Such deep convection can lead to thermocline convection affected by side topography of these basins (middle and southern basins). These deep topographically influenced rotating flows constituteparts of the abyssal circulation of the Caspian Sea.

The main aim of present work is to investigate the existence of deep abyssal overflow in the Caspian Sea. To fulfil this, we used observational data and numerical model showing that the overflow can exist in this basin. Firstly, we used observational data to understand the feasibility of deep flow in this basin, however, the resolution of observation data is very low to cover all the purpose of the paper. Hence, the use of a numerical model can help us for a better understanding of this deep overflow. Overall, this paper can be divided into three main parts based on the main goal of the paper. Section two focuses on the existence of deep flow in the Caspian Sea using some observations and numerical simulations. This section intends to answer as why the overflow can occur in this water basin as there are small research work on this deep flow in the Caspian Sea (Peeters et al, 2000). In the course of this section, the accuracy of the model results with observational data is investigated. Section three concentrates on the dynamics of the outflow when moving through the strait into the southern basin. Although there are many aspects to investigate the dynamic of the flow, the vorticity and potential vorticity will be considered in this section. Section four, the importance of the abyssal overflow will be indicated and the volume of the flow will be calculated using a simple formula. In the following this section, it will be clarified as why we follow this approach in this paper.

## 2 Data used and method of the research

2.1 Observational data

The CTD data are obtained from UNESCO Atomic Energy International Agency, for the summer of 1995 and 1996. This data is recorded in September (the same month for two years) and are collected in 42 stations using an exploration ship in this project namely Hajef. Among all stations, 13 and 29 stations are related to 1995 and 1996 periods respectively (Figure 1). In the first step, temperature, salinity, and density diagrams are plotted for all stations. Based on these data, the difference in density between the middle and southern basins is observable. For example, Fig. 2a shows the density differences between a and b (Fig. 1b). The density difference for these two points is estimated as 0.5 kg/ m$^3$. For better understanding, the *T-S* diagram is plotted to investigate the contribution of temperature and salinity in the density difference. Based on the *T-S* diagram, the water in the middle basin is both cooler and saltier than that of southern basin particularly in the deeper parts. Hence, there should be a deep abyssal overflow between two basins over the sill of the Strait separating them. The temperature, salinity and density transects across the Absheron Strait, as shown in figures 3a, b show evidence of this deep abyssal overflow moving from the middle to southern basin. As the sloped isopycnals are similar to isotherms, it seems that the buoyancy driving the flow is mainly due to temperature difference. However, unfortunately the horizontal resolution of the measured data is not good enough to show more sensible patterns of these parameters (see below).

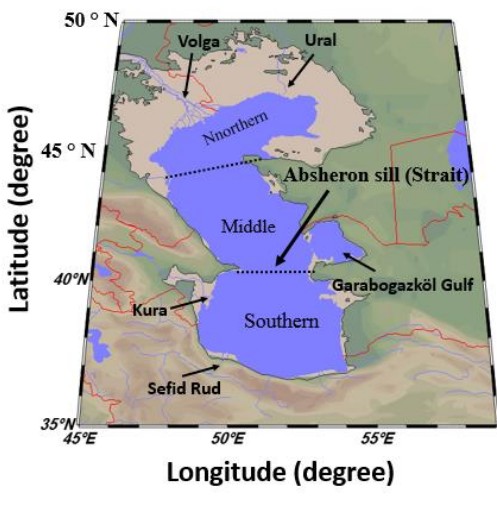

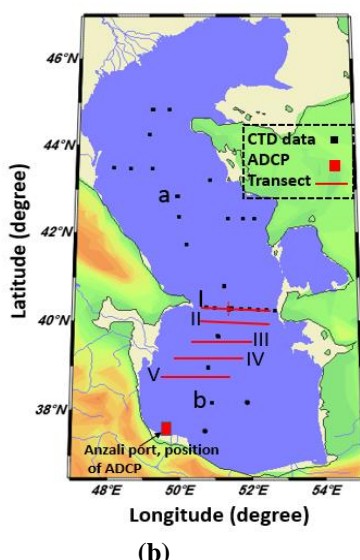

**(a)**                                                             **(b)**

5  **Fig 1:** (a) Schematic diagram of the Caspian Sea with the locations of the most important rivers indicating the Volga, Ural
and Kura and SefidRud, Garabogazköl Gulf and Absheron Strait are also showed on the map, (b) locations of CTD and
ADCP measurements, the geographic position of transects. The CTD casts are for 42 stations for September 1995 and
1996. The CTD stations, a and b are emphasized because in the Fig. 2a, b physical properties of the water for these points are
presented. ADCP data is recorded from November 2004 to the end of January 2005 to validate the Numerical simulations.

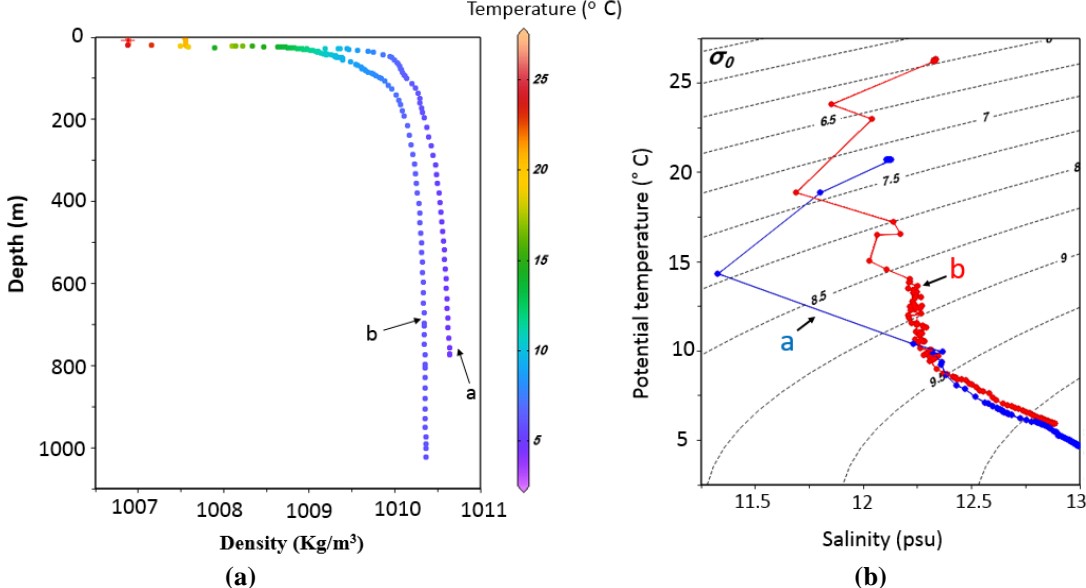

**(a)**                                                             **(b)**

**Fig. 2:** (a) Comparison of density between points a and b (middle and southern basins, see Fig.1b) indicating the difference
in density (~ 0.5 kg /m$^3$) between two basins. (b) *T-S* diagram for a and b to show differences in temperature and salinity,
15  particularly in deep water. To plot this diagram, the potential temperature and potential density anomaly ($\sigma_0$) are calculated
from CTD data. The *T-S* diagram confirms the differences in density in deep water for $\sigma_0 > 10$ kg /m$^3$.

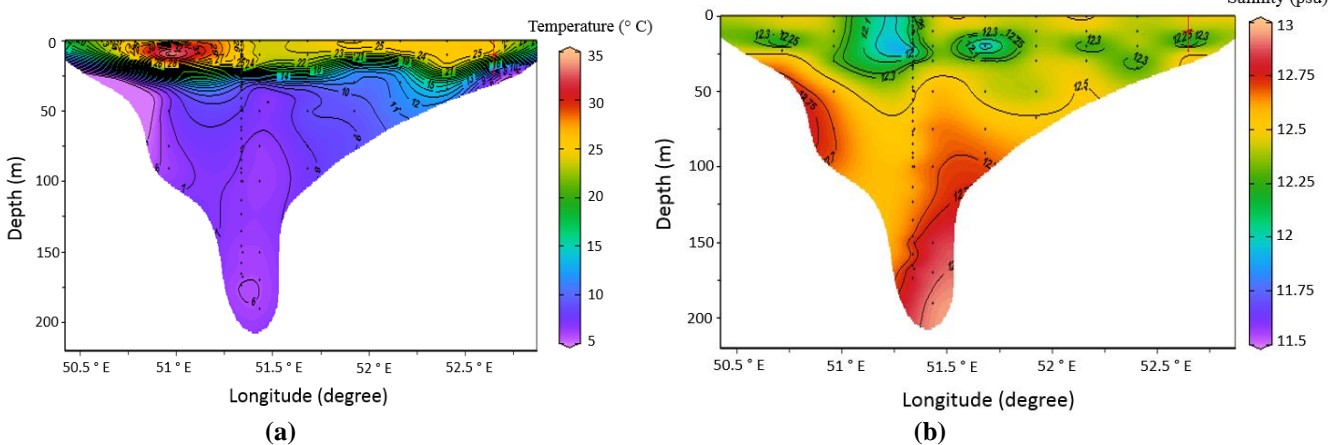

(a)                                                        (b)

**Fig. 3:** Cross sections of temperature and salinity across the Strait at transect I (across the Absheron Strait over the sill) in September from observational data; dots show the locations of measurements.

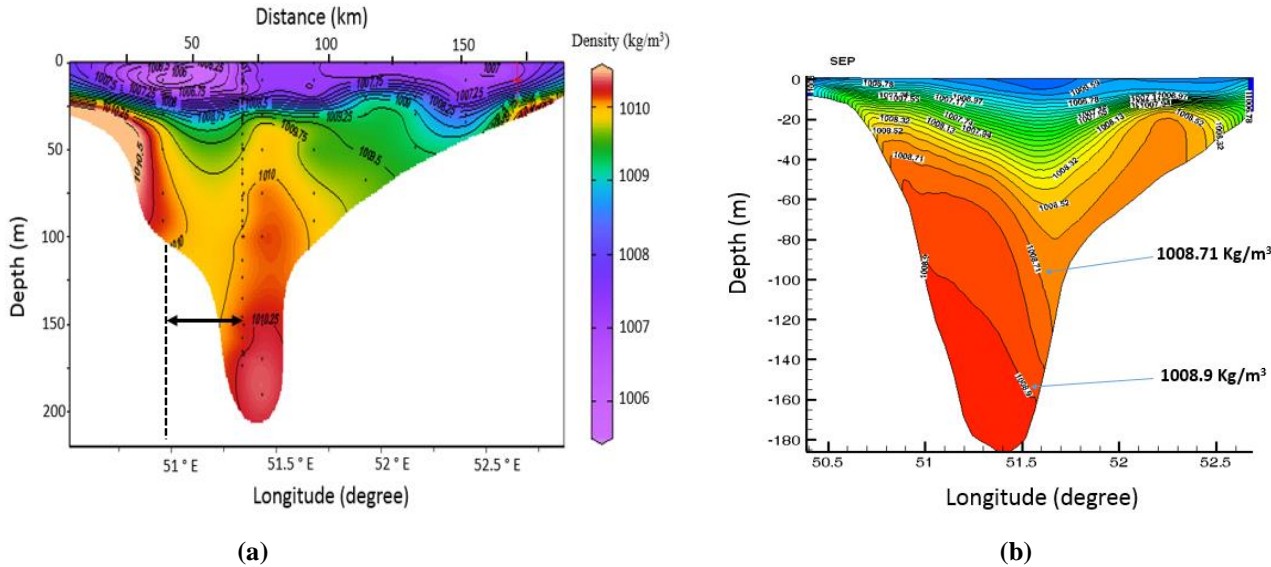

(a)                                                        (b)

**Fig.4:** Comparison between density fields across the Strait in transect I in September, from observational data (a) and numerical model (b). For observational data dots show the locations of measurements. For a better understanding of the spacing of CTD stations, distance is plotted in kilometres on the top of Fig. 4a. The cross sections are about the same but the differences between the two transects near the bottom, particularly in the west of the Straitis is mainly due to the low resolution of observational data (4a).Numerical transect clearly show the boundary trapped overflow for which two isopycnals near the bottom is highlighted (4b).

The horizontal resolution of observational data is very coarse for showing the overflow. For example, the Fig 4a indicates that the width of the Strait is about 200 km and we have just 9 CTD stations. It means that the average distance between two stations is 23 km, however, unfortunately in the most important area of the Strait (western) the distance between two stations is 30-35 km. As a result, we have some problems in showing the structure of the flow. More fine-resolution data and data for different

months are required to compare the cross sections of the flow for different months. Hence, due to lack of good measurements around the Strait between the two main basins of the Caspian Sea we are compelled to use numerical simulations for the study of deep overflow at the Absheron sill, including its seasonal variability. This also includes some general aspects of the circulations in the Caspian Sea. To achieve our goal, we use the numerical model and some field data.

## 2.2 Some general features of the COHERENS numerical model, and its boundary and initial conditions

For the simulations the numerical model COHERENS (Coupled Hydrodynamical Ecological model for Regional Shelf seas; Luyten et al., 1999) has been used. COHERENS uses a vertical sigma coordinate and the hydrostatic incompressible version of the Navier-Stokes equations with Boussinesq approximation and equations of temperature and salinity. The model uses an Arakawa C-grid (Arakawa and Suarez, 1983) and equations are solved numerically using the mode-splitting technique. The grid size in horizontal is $0.046 \times 0.046$ degrees, typically 5 km, and 30 sigma layers, labelled k (the bottom layer is 1 and the surface one is 30). The coastlines and bathymetry data with $0.5' \times 0.5'$ (30 seconds) resolution are acquired from GEBCO, although interpolated and smoothed slightly.

The model was initialized for winter (January) using monthly mean temperature and salinity climatology, obtained from Kara et al (2010); and it was forced by six hourly wind, acquired from ECMWF (Mazaheri et al., 2013), and air pressure and temperature with $0.5° \times 0.5°$ resolution acquired from ECMWF (ERA-Interim) reanalysis. Precipitation rate, cloud cover and relative humidity ($2.5° \times 2.5°$) were derived from NCEP/NCAR re-analysis data. The river inflows (from the Global Runoff Data Centre) were also included. The time steps of barotropic and baroclinic modes are 15 s and 150 s respectively. The total simulation time is five years (from 2000 to 2004 inclusive) with six-hour varying meteorological forcing and then the results of the last year are shown. The results of the numerical model are validated by ADCP data between the estuary of the Sefidrud River and Anzali port (figure 1, 5). These data are collected by National Institute of Oceanography and Atmospheric Sciences, from November 2004 to the end of January 2005 (Shiea et al., 2016).This data was recorded by RCM9 current meters (at the ADCP station) at 3 depths on a mooring, near the surface, 50 m and 200 m. The lack of observation data is the main obstacle to check the accuracy of model. It would be more useful to have data on the sill for the model validation. However, using ADCP data near the Iranian coast was our only option to fulfill this. For this reason, the model is applied for these years because it can be validated with ADCP data in some months of last year of model runs.

The simulation results of mean and long period variations of surface velocity components are rather consistent with observations. This similarity relates to timing of flow variations rather the velocity magnitude.

The difference in velocity between observation and model simulations comes from some of the assumptions and the resolution used in the model as can be expected. The distance between two adjacent grid points in the model is about 5 km and the ADCP data are a point in between two grid points, so interpolation is used to compare the model results with data at the location of observations.

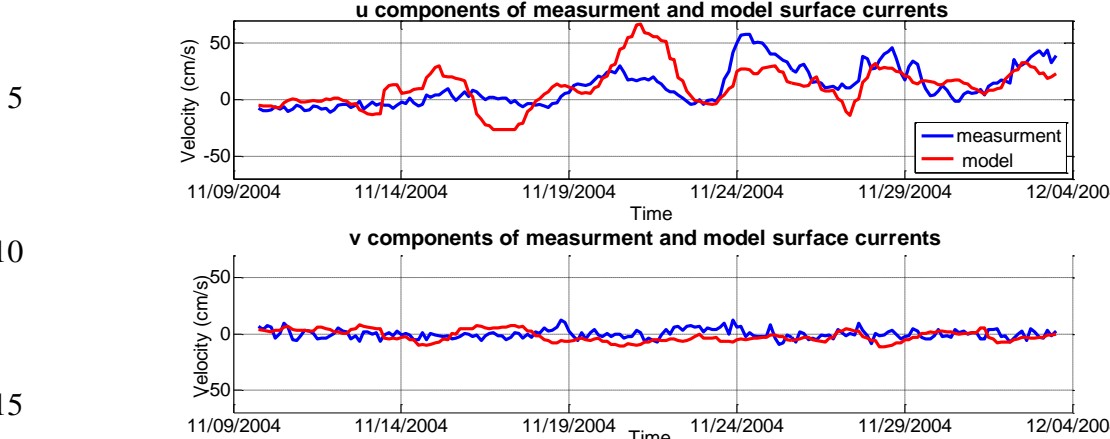

**Fig. 5**: Comparison between numerical model results of surface current components and observation near the Sefidrood River and Anzali port (Shiea et al., 2016).

At the beginning of this paper, observational data indicated that the deep flow could exist on the sill (Fig.4a). The numerical simulations also show that the deep flow clearly exist on the sill (Fig. 4b and 6), which we examine here with more details. Typical numerical results of deep flows for the middle and southern basins of the Caspian Sea (the flow in the northern basin is not shown as it is too shallow) for May and December of 2004, after four years of warm-up of the model, are shown in Figure 6, 7, 8, and 9. The deep narrow flow in the middle basin and the overflow over the Absheron sill and in the north western boundary of the southern basin are clearly observed.

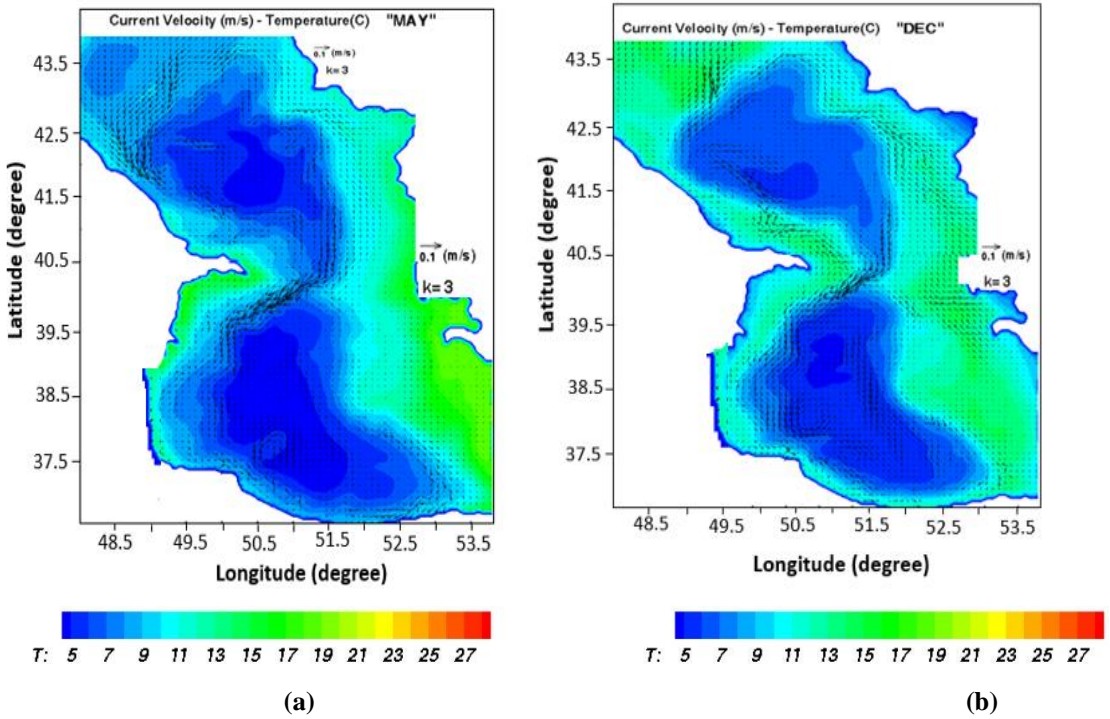

**(a)** **(b)**

**Figure 6:** Monthly mean currents (m/s) in layer "k=3" (near the bottom) and temperature obtained from model simulations in southern and middle basins of the Caspian Sea for the months of May(a), December(b) (2004).

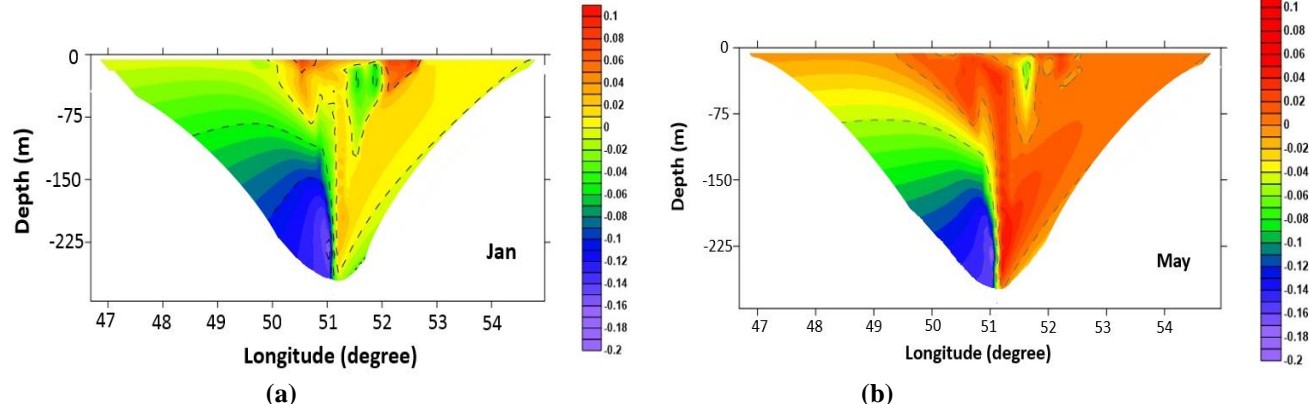

**(a)** **(b)**

**Figure7:** Cross-Section of the mean velocity (m/s) in transect I obtained from model simulations, (a) for January and (b) May.

## 2.3 Comparison of numerical simulations with observations

The main reason for the deep flow existence in the Caspian Sea seems to be the temperature differences between the northern and southern basins. The sea surface temperature (SST) in the northern basin ranges from below zero under frozen ice in winter to 25–26 °C in summer, while more moderate variability occurs in the southern basin changing from 7–10 °C in winter to 25°–29 °C in summer (Ibrayev et al., 2010). It shows that the water in the northern basin cools in the cold seasons so that it freezes. On the other hand, the Caspian Sea has low salinity and in deep areas, salinity varies little with depth (12.80–13.08 psu), so that the density stratification largely depends on temperature variation (Terziev et al., 1992). As the northern shallow waters of the Caspian Sea are subjected to high evaporation in summer, in the following cold seasons these waters become dense and start to sink, mainly in the northeastern side of this sea (Gunduz and Özsoy, 2014. Based on the present work, the flow due to its high density enters the deep part of the middle Caspian and starts to fill the middle Caspian Sea basin. After filling the middle Caspian basin, it appears as an overflow entering into the southern basin (10a), due to the shape of Absheron Strait which is similar to the that of the Denmark Strait (DS) sill. Although this overflow is smaller than that of the DS.

In order to compare the numerical simulations with observational data, some vertical distributions of temperature, density are presented (Figs.4, 8). Figure 4 indicates that the numerical model simulates the density lower than the real value on the Strait. This difference is about 0.5-1 kg/m³ from the surface to bottom with more difference in deep part. However, we can observe the similarity in the shape of the isopycnal lines between the numerical model and observations particularly in the eastern part of the Strait. These difference sare related to our assumptions and simplifications in the numerical model. We do not consider the Garabogazköl Gulf which can be an important factor for producing higher salinity water in the middle Caspian Sea due to high evaporation in this area (see Fig 1a). Over the years, the waterway connecting the Garabogazköl Gulf to the Caspian Sea is open for some years or closed for the other years based on the fluctuation of sea surface level in the Caspian Sea. However, accurate information about the connection is not accessible for whether to include the higher-salinity source in the numerical model simulations. As a result, this factor can be important in underestimation of density by the numerical model. Apart from this, the comparison of temperature between the results of the numerical and observational data indicates that the numerical model shows a higher temperature than that of observation data at the same depth (Fig. 8 a, b). For example, if we consider the isothermal line for the potential temperature of 6 degrees Celsius (see Fig. 8 a,), it is at a depth of 200-300 in the middle basin in the numerical simulations, while this isotherm is at about 200-250 m in observational data. As a result, the numerical model calculates the density less than its actual value. Based on what was mentioned, two factors contribute to the formation of deep flow between the middle and southern Caspian Sea basins. We generally conclude that the temperature factor in the formation of this deep flow is more important, because the isopycnals are very similar to isotherms over the Strait (see 3a, b and 4a). Some other works also confirm the importance of temperature in the structure of circulation of water in the Caspian Sea (e.g. Terziev et al., 1992) and Ibrayev et al., 2010).

Typical Rossby number of the overflow is about $Ro = \frac{U}{fW} = 0.2/(10^{-4} \times 20 \times 10^3)$ ~0.1 (here U is typical speed of the overflow and W is its width), which justifies the geostrophic flow (assumption) entering the southern basin.

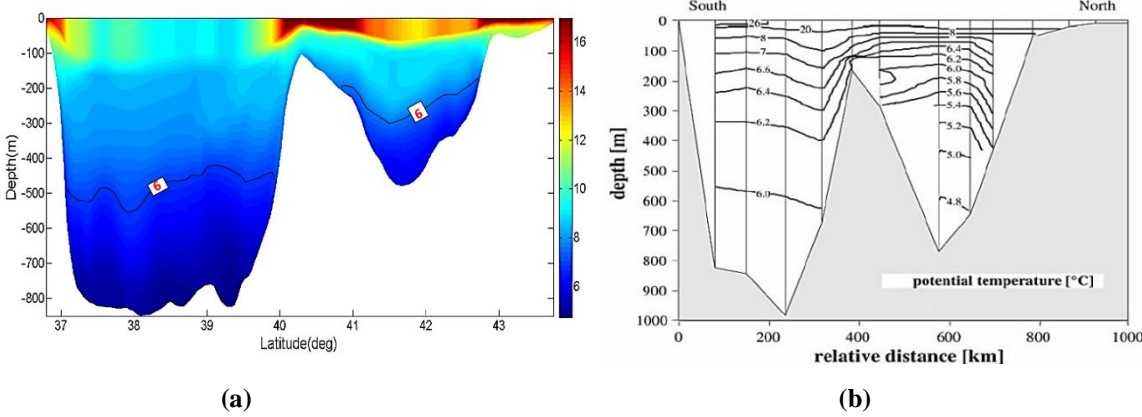

**(a)**                                                        **(b)**

**Figure 8:** Comparison between the vertical cross-sections (along the north-south, N-S) of the mean temperature obtained from model simulation (a) and Peeters et al. (2000) measurements (b) during September. The 6 °C isotherm is marked for easier comparison.

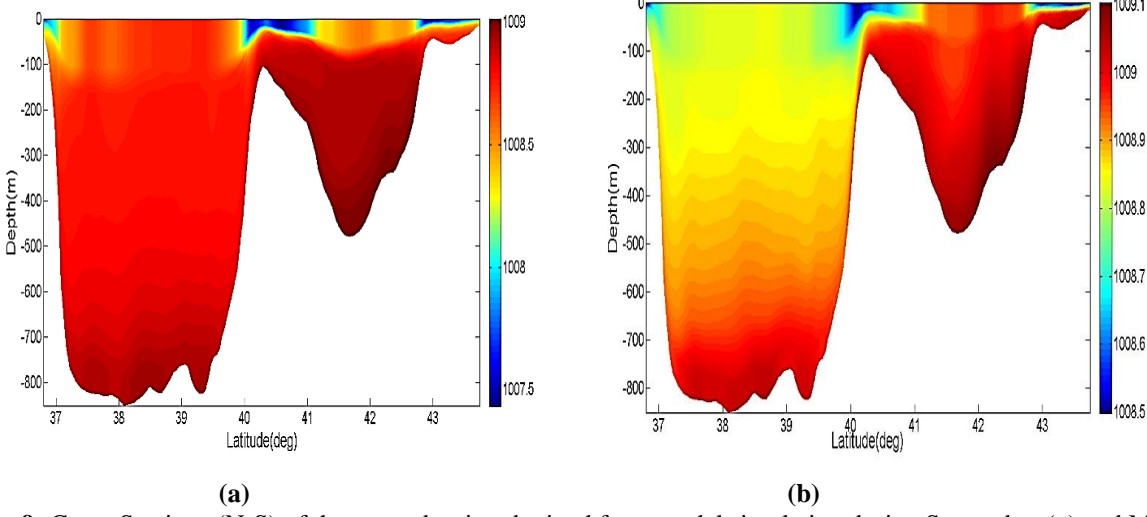

**(a)**                                                        **(b)**

**Figure 9:** Cross-Sections (N-S) of the mean density obtained from model simulation during September (a) and May (b).

Based on the numerical model, main physical properties of water are changeable in different seasons. For this, some of the variability features are show in table 1. $\lambda$ is the flow angle from zonal east-west direction (please see Fig. 10 a). Due to various initial conditions for deep flow, it probably leads to different behaviors in southern basin.

**Table1:** Boundary current parameters and variables obtained from the numerical simulations on the sill based on transect I (see text).

|  | $g'$ (m/s$^2$) | $v_p$(m/s) | $u_p$(m/s) | $\lambda$(deg) |
|---|---|---|---|---|
| NOV | 0.00222 | -0.12 | -0.044 | 107 |
| JAN | 0.00239 | -0.133 | -0.047 | 109 |
| MAY | 0.00251 | -0.147 | -0.031 | 101 |
| SEP | 0.00241 | -0.19 | -0.078 | 112 |

What stands out from table1 this is that the main features of deep flow are different in each season. These differences show that the deep flow velocity and reduced gravity fluctuates during the year, 0.127-0.2m/s (magnitude of the velocity components) and 0.00221-0.00251m/s$^2$ (reduced gravity) respectively. As a result $\lambda$ is changeable from 101 to 112 degree. It is predicted that the flow may show different behavior when moving over the Apsheron sill and then into the southern basin.

Here, we focus on the how much water sinks with different initial conditions on the strait. To fulfill this, some transects are plotted and shown in Fig.11. These transects are related to I and II (see Fig. 1b) to evaluate how much the water sink after moving 20 km. As examples of months, January and September and the isopycnal 1008.9 kg/m$^3$ are chosen. This choice comes from consideration that the isopycnal lines should have similar patterns. For example, maximum isopycnal in November is 1008.8 kg/m$^3$ which is difficult choose for a better comparison because the same isopycnal should be used for all months. I

and II are used due to the fact that there is a little distance between them and because the entrainment and friction effects are less on the overflow rather than some transects like IV.

In this section, it is tried to investigate the effects of the different initial conditions. Based on the results of Fig.11, similar isopycnals are located at depths 105 m 125 m for January and September respectively. This depth is associated with the mean values of the maximum and minimum depths where the isopycnal is located. The results indicate that the deep flow in summer

is more near to the bottom than in winter. In general, the mechanism of formation of water mass can be very complex. This difficulty is related to the formation time of this water mass and how long it takes that the water reaches the Strait. To clarify this, in the previous section we mentioned that the dense flow fills the middle basin and then overflow in the southern basin (10a). For this reason, it is tried to estimate the filing time in section four with the acceptance of some assumptions. However, Peeters et al (2000) estimated this value about 20-25 years. Because of this, the density of water entering the southern basin is

not the same as the water sinking in the northern basin only in winter. Nevertheless, it would be possible to track the sinking water in the strait if the numerical model run were at least for 20 years. The present numerical model runs are for four years only due to computing limitations. If the model runs were extended for many years, it would be likely that the waters that were in the early years of the running model had slinked, and could be seen in the next few years on the Apsheron Strait. In addition to this, the spectral density of the vertical water velocity variation that sunk in the northern basin and the velocity changes in

the flow that enters the southern Caspian water can be plotted; it is likely that time scales of variability of the dense flow on

the strait can be better investigated. It could also be concluded in which years the outflow is stronger or weaker and also this can another method to calculate filling time of the basin.

 If we accept these points, a question may be raised in the mind as why these simulations still show the deep flow, despite the fact that the simulation time was shorter (4 years), rather than time of formation and filling of the middle basin which is 20

years (Peeters, et al, 2000). It should be noted that the initial condition of the present numerical model is the outputs of the HYCOM model which was run by Kara et al. (2010), which has the proper density field after a long run. It means that if a numerical model runs for the Caspian Sea was with any initial condition (e. g. uniform waters temperature and salinity), it would take at least 20 years to show a deep flow. It is very common in running a numerical model using the same temperature and salinity with depth from the surface to bottom when there are not any observational data. In the course of that, the model

needs more time to be stable. This may be an important reason why other previous numerical simulations did not show any deep flow, as the years of running the models were not enough for the basins to reach a quasi-steady state.

By comparison of transects I and II, it is indicated that the water sinks about 200 and 80 meters in September and January respectively, as the overflow enters the southern basin. This occurs when the water moves nearly 23 km (the distance between I and II) along a latitude line. One of the most important reasons for its depth variation can be difference in reduced gravities

that varies with season.

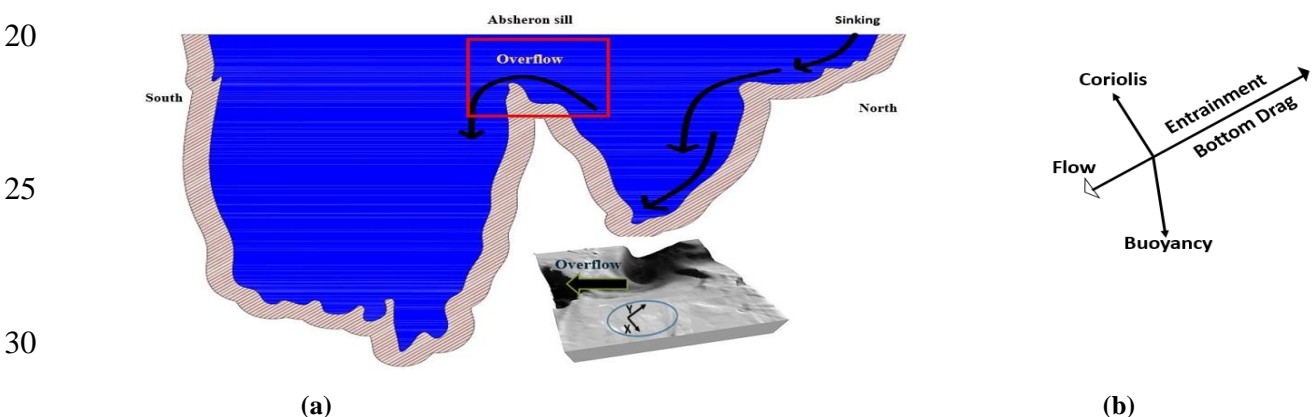

(a)                                                                (b)

**Figure10:** (a)A schematic diagram of the sinking flow in the middle basin and the overflow current over Absheron sill

(top),and topography around the sill in the middle of the Caspian Sea with the chosen coordinates (bottom).(b)Balance of forces on the overflow and flow coordinates are also shown.

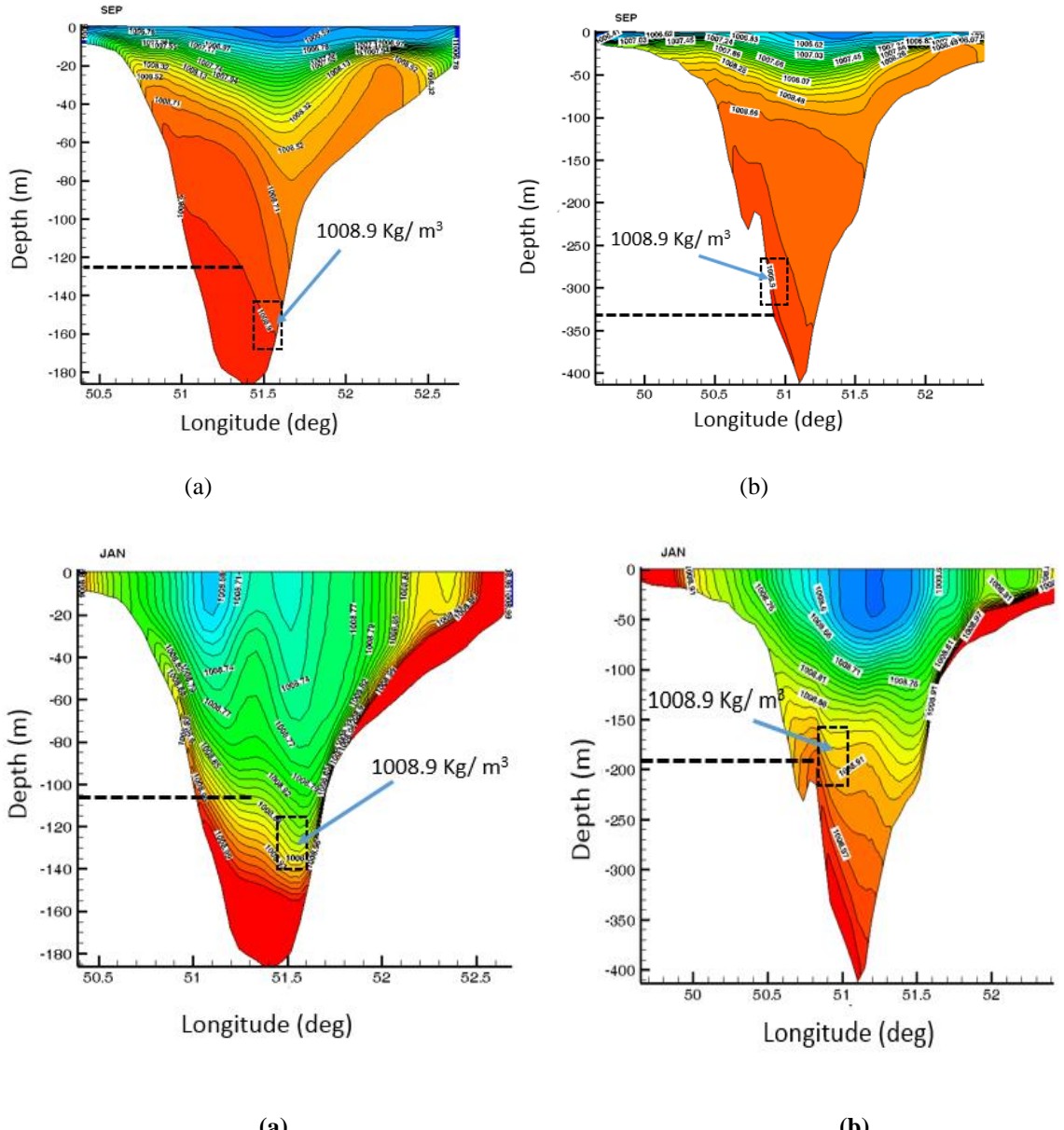

**(a)**                **(b)**

**Fig 11**: Simulated density fields along transects I(a) and II (b) in Sept. (above) and January (below) 2004.

## 3 Dynamics of the overflow

The dynamics of the flow can be analyzed using analytical models as have been used in other works. Among them, one can refer to Girton et al. (2003) work in which they used a stream tube framework to analyze the results of their observational data in Denmark Strait Overflow. For this reason, in this section it is tried to investigate the dynamics of the overflow in the Caspian Sea. The flow, after entering in the southern basin deflects to the right and being trapped on the southern basin topography. In the course of that, Coriolis, buoyancy, bottom frictions, and entrainment are the most important forces which affect the deep flow (Fig 10b). A better approach would be to use a method which includes all forces affecting it dynamically. Although there are many quantities which are important in terms of dynamics, the vorticity and potential vorticity are often used to investigate the flow behavior. Generally, the vorticity is one of the most important parameter in the oceanography for understanding the main features of water column when moving over a strait. It also clear that some of the main features come from the numerical model simulation results.

Based on what was mentioned in section 2, although the main aim of doing numerical simulation was to show the deep outflow in the Caspian Sea, in addition to this the outputs showed the new finding about the behavior of the flow in southern basin. The flow after being trapped in the southern basin create eddies, particularly near of the Iranian coast. As the flow reaches the SefidRud cape it separates from that and form one or two eddies. Similar to this behavior near the cape, the Persian Gulf outflow separate from the near the Ras Al Hamra Cape while being attached or detached from the Cape which depends on the outflow properties which vary with seasons (Ezam et al., 2012). For this reason, this section focuses on the vorticity and potential vorticity because the previous work showed the there is a link between the vorticity of the water column in upstream and separation of the flow from the cape. Due to the fact that the numerical model cannot directly calculate the vorticity and potential vorticity, we should utilize a formula to calculate vorticity alongside using the numerical simulations outputs. In addition to this, Falcini and Salusti (2015) presented a new method to estimate the vorticity of water column. This formula is very useful due to the consideration all of forces which are import in the present overflow dynamics. Thus, this method is used in this section. Here, firstly entrainment parameter and drag coefficient are calculated and then the dynamics of model is discussed.

## 3.1 Estimation of drag coefficient and entrainment parameter

Johnson and Sanford (1992) estimated drag coefficient, $C_d$=3×10$^{-3}$ from the analysis of data from the Mediterranean outflow. Girton and Sanford (2003) used $C_d$=3×10$^{-3}$ for the Denmark Strait and Cheng et al (1999) studied the bottom roughness length and bottom shear stress in South San Francisco Bay and calculated $C_d$ from 2×10$^{-3}$ to 6×10$^{-3}$. In this study, we have conducted the analysis using $C_d$=3×10$^{-3}$ and 5×10$^{-3}$. Hence, $r_b$ =$C_d$ $u$ /$H$=0.003×0.2/50 ~ 1 ×10$^{-5}$s$^{-1}$ and $r_b$= $C_d u/H$=0.005×0.2/50 ~ 2×10$^{-5}$s$^{-1}$. $Ri$ is the bulk Richardson number defined as $Ri = \frac{g'H}{U^2}cos\theta$, with $U$ being the amplitude of layer velocity, $\theta$ is the bottom slope. There are many methods to calculate entrainment parameter, $E^*$ of which some are presented in table 2. Due to the

importance of $E^*$ in the next section for estimation of vorticity, the $E^*$ is calculated based on table 2 formulas, for the transects I, II, III, IV, and V. Figure 12 plots $E^*$ versus $Ri$ for May from transects I to V.

Based on $Ri$ for the overflow, $Ri$ varies at different locations. Using $U$ as 0.1 to 0.2 m/s, $g'$=0.00222-0.00251m/s$^2$, $H$=50-70 m, $\tan\theta$=0.02, $Ri = \frac{0.00251\times50}{0.2\times0.2} 0.99 \sim 3.1$. Based on table 2 with Ri>= 0.8, we used mean $E^*$ based on formulas 2, 3, 4 and 5 of table 1, because we cannot use the formula 1($Ri$>= 0.8, then $E^*$<=0). For this section, $r_e$ values are considered as $5\times10^{-6}$ and $1\times10^{-5}$s$^{-1}$ based on typical values for $Ri$, $U$ and $H$.

**Table 2:** Some of the published $E^*$ equations based on $Ri$ (Kashefipour et al, 2010).

| Equation number | Researcher | Year | Equation |
|---|---|---|---|
| 1 | Ellison and Turner | 1959 | $E^*=\dfrac{0.08-0.1Ri}{1+5Ri}$ |
| 2 | Ashida and Shinzi | 1977 | $E^*=0.0015Ri^{-1}$ |
| 3 | Garcia | 1985 | $E^*=\dfrac{0.075}{(1+718Ri^{2.4})^{0.5}}$ |
| 4 | Kessel and Krancnburg | 1996 | $E^*=\dfrac{5.5\times10^{-3}}{3.6Ri-1+\sqrt{(3.6Ri-1)^2+0.15}}$ |
| 5 | Karamzade | 2004 | $E^*=0.0021\ Ri^{-1.1238}$ |

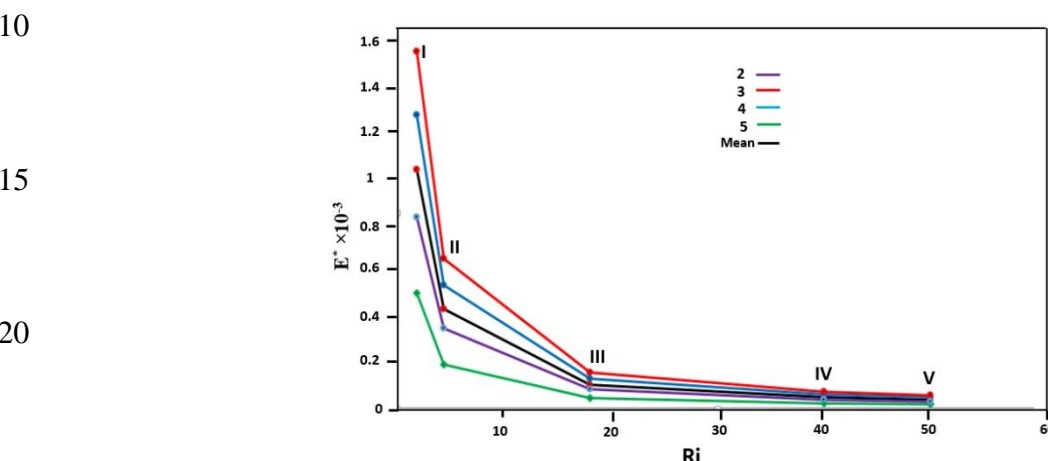

**Figure 12:** Changes of entrainment coefficients based on different Richardson numbers for May in transects shown in figure 1b. The formulas in table 1 are used to estimate $E^*$ (the numbers refer to the equation numbers in table 1). To use $E^*$ in Eq. 5and 6(section 3.2), mean values for $E^*$ is also estimated (black).

## 3.2 The changes of vorticity and potential vorticity of the overflow

Based on what was mentioned in the beginning of this section, vorticity is one of the other important parameter that is significant to the study the column properties of overflow over the sill. Apart from this, the vorticity is very useful to investigate

the behaviour of the flow (e.g. Stern, 1980) in the southern basin, particularly near the SefidRud Cape. Not only do we try to estimate the vorticity of the water column and the width of flow when moving into the Southern basin, especially the adjustment of the flow width, but also we estimate the flow vorticity and its behaviour near the Cape. The width of the flow is calculated directly from the numerical model but for the calculations of vorticity and PV (potential vorticity) we need to use an analytical model.

Here we consider the structure of fluid flowing over the sill in terms of its vorticity and PV. Falcini and Salusti (2015) presented an analytic model for the Sicily channel: vorticity and PV equations are based on the stream-tube model (Smith, 1975; Killworth, 1977). To deal with this, they used $(\xi, \psi)$ coordinate system, a modified form of that was used by Astraldi et al. (2001). In this frame, $\xi$ is the along-flow coordinate and $\psi$ is the cross-flow coordinate (see Figure 1 in Smith, 1975). In this method, friction and mixing effects are considered for estimation of potential vorticity, whereas other models assume a zero-

PV change for the flow (Whitehead et al., 1974). Falcini and Salusti used the hydrostatic pressure equation for third layer (with dense water near the bottom); while considering a third denser layer, they used the depth change of the flow layer to account for the effect of upper layer, rather than depth change of the dense layer. In addition to this, based on their assumption (Falcini and Salusti, 2015), the velocity of a stream line is a function of $\xi$ only. They defined $\beta$ as the angle between $(\xi, \psi)$ and $(x, y)$ coordinates. They assume that $\beta$ is close to zero in the channel. They used the classical vorticity equation (Gill, 1984) and

assumed cross sectional averages of the various terms in the steady state of classical vorticity equation due to difficulty of depth and velocity calculations in different position from hydrographic data. They presented two formulas to calculate vorticity and potential vorticity (1and 2). Formulas are based on a homogeneous bottom water vein and using shallow water theory (over bars indicate cross sectional averages). To obtain a formula, the bottom water is assumed to be well mixed and the flow has a strong axial velocity, nearly uniform over the cross section of the stream and also the cross-stream scale is assumed to

be much smaller than the local radius of curvature of the streamline axis. Relative vorticity and potential vorticity distributions of the flow are:

$$\frac{\bar{\zeta}}{f} = \frac{\bar{u}_0}{\bar{u}} e^{-\int_0^\xi \frac{r}{\bar{u}} dx} \left( \frac{\bar{\zeta}_0}{f} + \frac{1}{\bar{u}_0} \int_0^\xi e^{\int_0^x \frac{r}{\bar{u}} dx'} \left[ \frac{\bar{u}}{\bar{h}} \frac{\partial \bar{h}}{dx} - \frac{E}{\bar{h}} \right] dx \right) \tag{1}$$


$$\Pi = e^{-\int_0^\xi \frac{\Gamma}{u} dx} \left[ \Pi_0 - \int_0^\xi e^{\int_0^x \frac{\Gamma}{u} dx'} \frac{r\zeta}{hu} dx \right] \tag{2}$$

Where $\Gamma=\dfrac{E^*}{h}$ .

To obtain Eq.1, it is supposed that $\overline{\zeta_0} \ll f$ . After integrating the shallow-water equations along the flow and mass continuity equation and also using some mathematical operations, Eqs. 1 and 2 are obtained.

Here $\zeta$ and $\Pi$ are respectively the mean relative vorticity and potential vorticity, $h$ is the layer thickness and $\partial h/\partial x$ is slopes of the isopycnals, $\zeta_0$ and $\overline{u_0}$ are respectively the initial vorticity and velocity. $\zeta_0$ is estimated as $U/W$, where $U$ and $W$ are respectively speed ($\sim$ 0.2 m/s) on the sill and cross channel scale ($\sim$ 20 km) over the sill. To be applicable, some terms in Eq.1are considered as cross-sectional averages. Three terms are significant in vorticity: a stretching term, the entrainment effect, and friction. In 1and 2, some symbols have been changed, compared to the present work. To estimate all parameters in these formulas, we use 5 transects from Strait (I) to the southern area (V) (figure 1b).Typically, for November in transect III, $\partial h/\partial x\approx0.0047$ ($\delta h\sim 180\ m\ and\ \delta x\sim 38$ km), $\bar{u}=0.11$ m/s, $\bar{h}=180$ m, are estimated. In the calculation of $r_b$, based on $Ri$ number $r_b=2\times10^{-5}$ (s$^{-1}$) and $r_e=4\times10^{-6}$(for transect II), $8\times10^{-7}$(for transect III) are calculated based on formulas in table1 (figure 12). In transect IV and V, $r_e\sim0$ because of large $Ri\sim$ 40 to 50. Using Eqs. 1and 2, profiles of $\zeta$ and $\Pi$ are plotted in figures 13as functions of $\xi$ along the steam tube.

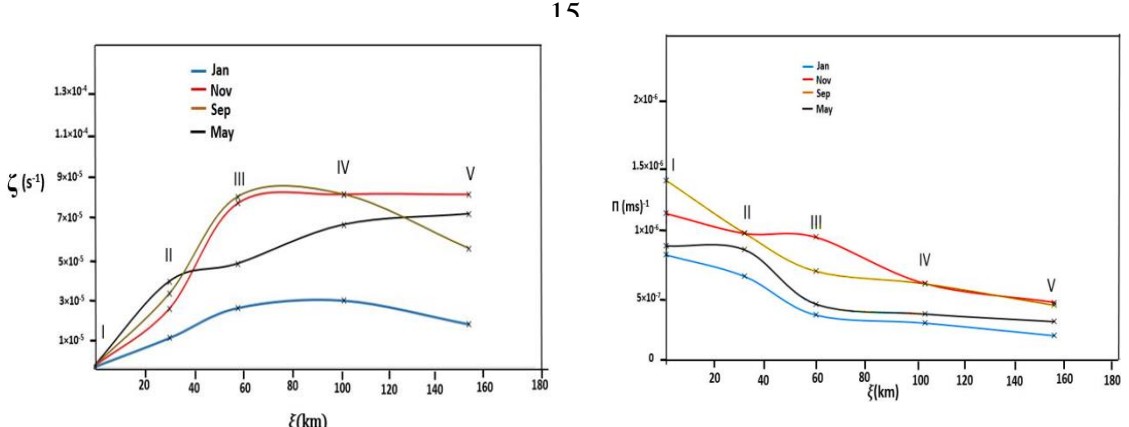

**Figure 13:** Changes in $\zeta$(s$^{-1}$) (left) and $\Pi$ (m$^{-1}$s$^{-1}$) (right) along the flow based on Eqs. 1 and 2.

Figure 13shows that $\zeta$ increases from I to V, because of stretching term in Eq. 1, although Sep. and Jan. values have different behaviour in transect IV to V. However, after the transect III, the changes are not considerable, because the depth does not vary significantly (stretching term) and entrainment has been ignored as $Ri$ is large after this transect. In the month of November, the vorticity has maximum value of about $8.5\times10^{-5}$ (s$^{-1}$) in transect V, although in January the vorticity is minimum among other months vorticities and also the changes are very little after transect III. When it comes to $\Pi$, the graph shows decreases of PV values along the flow from I to IV, but after IV, the $\Pi$ values are almost constant. For example, changes of $\Pi$

over the sill (from I to III) are about $7\times 10^{-7}$ $(m^{-1}s^{-1})$ and $4 \times10^{-7}(m^{-1}s^{-1})$ for Sep. and Jan. respectively, due to bottom friction and entrainment.

The transects I, II, and III are located on the slope and IV and V are in deeper parts of the southern basin. Although this conclusion is based on the topography of the Caspian Sea, the Fig. 13 shows an important point in that the changes of vorticity are more intense from I to III because the depth of water are changing more on the slope (stretching term) over this distance. As the flow enters the southern Caspian Sea basin, it adjusts into an internal Quasi-geostrophic flow, almost as a deep western boundary current in the southern Caspian basin. The gravity flow appears as trapped current after Absheron sill due to Coriolis effect (transect IV and V).When moving along the southern (Iranian) coast, the forces of pressure gradient and Coriolis balance the force of friction. The entrainment effect can be ignored because the Richardson number is about 50 based on Fig. 12 in transect V so $E^*\sim 0$. The width of flow when being trapped is calculated for all month. The width of flow is different for various seasons based on transect V. These values are 18, 16, 34, and 35 km for Nov, Jan, May, and Sep respectively. The width of the flow over the sill and in the southern basin is directly calculated from the numerical model simulation results. As figure 13 shows, the potential vorticity of the water column decreases from I to V, due to friction and entrainment. The comparison of the flow width from I to V transects also shows that the bottom friction (and also entrainment particularly on the Strait) increases the width of the flow in the southern basin. For example, for September, the width of flow increases from 20(I) to 35(V), by about 15 km as a result of moving over the sill and into the southern basin. It means that friction force decreases potential vorticity of the flow in the southern basin.

Apart from this, the trapped current continues its movement into the western part of the southern basin (Fig. 16), but it shows an interesting behavior as it reaches the Delta SefidRud cape. The flow separates from the cape and forms one or two eddies (Fig 17). Based on the numerical results, separation of the dense flow from the cape depends on the season (different boundary currents). The important parameter determining the behavior of flow when it separate from cape is its potential vorticity.

In this section, the potential vorticity of flow is estimated for different seasons based on information as in figure 15.We can observe different behavior of the flow when separating from the cape for fall in November and in spring May (Fig.15). For example, in transect V for November and May the values of the potential vorticity are $6\times 10^{-7}$ and $4 \times10^{-7}$ $(ms)^{-1}$respectively (Fig.13). Figure 15 indicate that in November, the flow is closer to the cape rather than in May during the time of separation. It can probably be concluded that the potential vorticity upstream of the flow can be effective on the flow when it separates from the cape, although other factors such as Rossby number is also important. In order to be more accurate, Stern (1980) showed that for this kind of the flow with zero potential vorticity assumption, the flow separates from the cape when the width of flow upstream of the cape is less than about $0.42$ $R_D$, where $R_D$ is the Rossby radius of deformation (based on the current depth, $H$ and its reduced gravity, $g'$ far upstream, given as $R_D= (g'H)^{0.5}/f$). Based on Fig. 13, we can still use the Stern method for this flow, although the potential vorticity is not quite zero upstream of the Cape (figure 13, right). Based on typical values of the $R_D\sim 2L$which is about 30 km, and width of the western boundary current (about 15 to 35 km, calculated above) which is in the same order as $0.42R_D$, the flow may just be separated from the cape especially for Jan and November (with width 16 and 18 km respectively), as indicated in Fig. 15 in which the separation and formation of a cyclonic mesoscale eddy near the

cape is more pronounced in November, considering the fact that PV of the flow is not quite zero before the cape, as the Stern criteria may not apply for the such flow separation. A more rigorous criterion is needed for the separation of such flow from the cape that may be dependent of the geometrical dimensions of the cape as well.

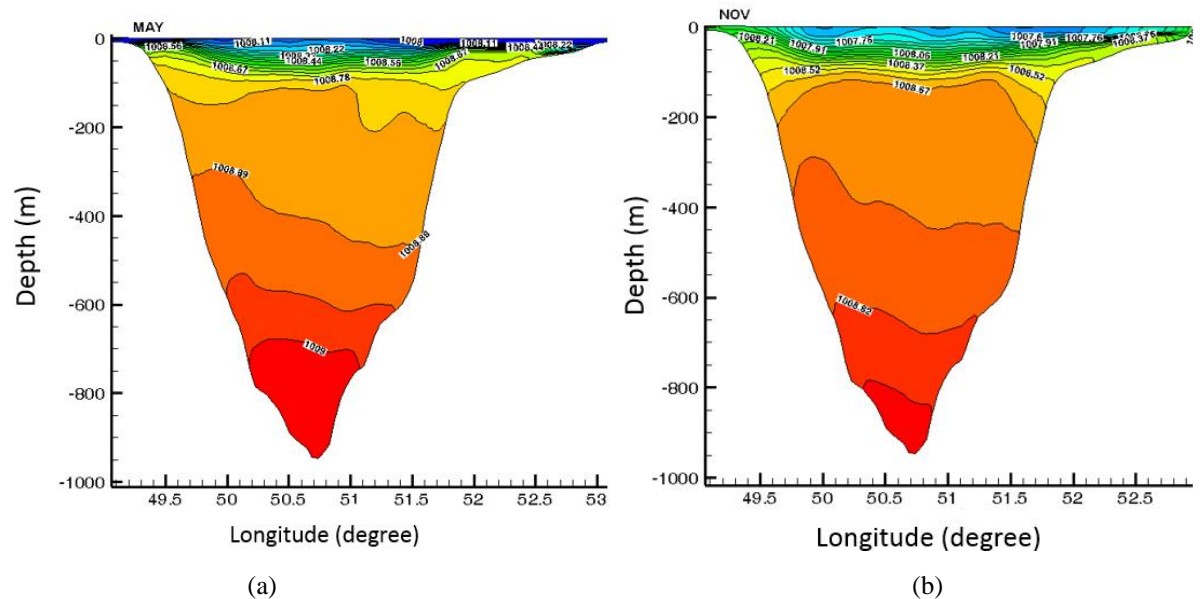

(a)                                                        (b)

**Figure14**: Density fields along transect V (from the numerical model) in May (a) and Nov. (b).

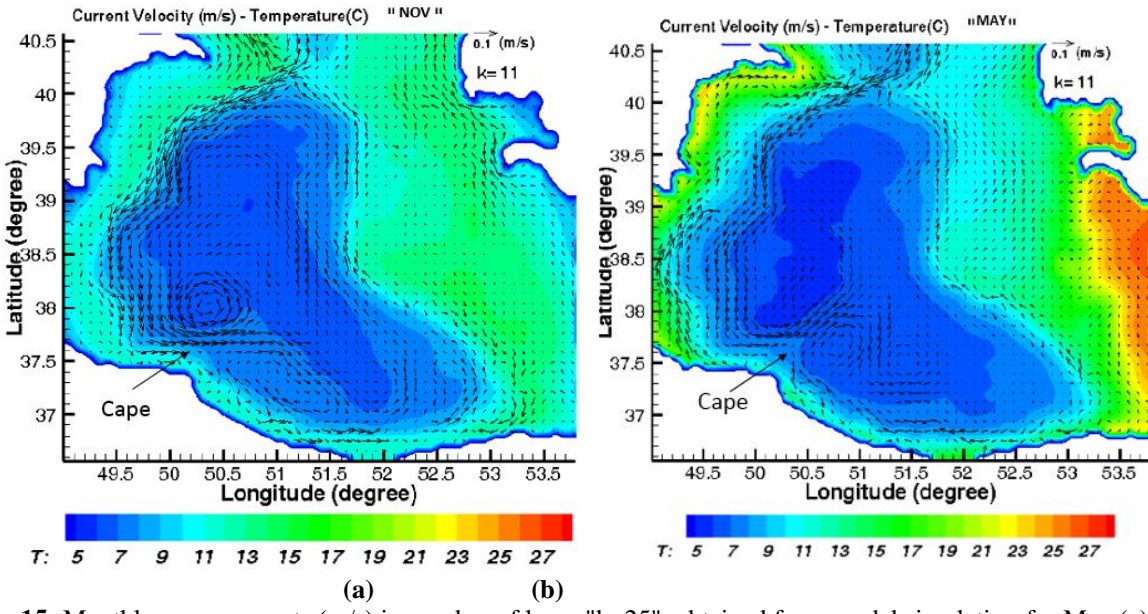

**(a)**                        **(b)**

**Fig.15:** Monthly mean currents (m/s) in number of layer "k=25", obtained from model simulation for May (a) and Nov (b). The dense flow separates from the SefidRud Delta cape.

## 4. Flushing time and the importance of this work

### 4.1 Volumes of basins dense water and flushing times calculations

Based on what was discussed in section 2 and also will be discussed in the following section, the flushing time is an important quantity for these basins. To calculate this, the first and important step would be the estimation of the dense flow volume flux when entering the southern basin on the Apsheron sill. A simple method to calculate it is multiplying the mean velocity of the overflow by its cross-section (Eq. 3). Although the numerical model outputs give directly the velocity, the cross- section should be calculated. It means that similar to section 3.2 when vorticity and potential vorticity are calculated with the formula (Falcini and Salusti, 2015), the volume is estimated by a formula. This is the main reason to present a formula, in addition to the numerical model outputs. To tackle this, it is very useful to use an equation which is compatible with the physical condition of the Apsheron sill. For this reason, the shape of the sill and the isopycnal will be considered for obtaining this formula. The most important point to present an equation would be the accuracy of a formula. Here the accuracy of this is checked by the simple method using the numerical simulations. Although using the observational data is more common for this type of research, the consideration of some points is also important. Unlike the work of others, here if we had ADCP data, we should have used the same method to calculate the overflow volumes flux, which means that converting the cross-section to different grids and obtaining the result of the multiplication of flow velocity by the grid area. To be more applicable, it is tried to obtain a formula under conditions without any need to ADCP data. In general, the temperature and salinity data (CTD) are much more than the ADCP data along the straits like the Apsheron one, and also the Hormoz strait. Thus, we follow this method to estimate volume flux on the strait.

The overflow volume flux is given by (3) in which $v$ is the mean magnitude of geostrophic velocity of the overflow and $ds$ is an element of its cross-section area.

$$Q_V = \int v ds \qquad (3)$$

Due to the parabolic form of the bottom topography of Absheron Strait, its geometry of the dense overflow in this valley like shape (Figure16a,b) can be given by:

$$Z = ax^2 + bx + c$$
$$h' = Ae^{-\alpha x} \qquad (4)$$

Where $a$, $b$, $c$, $A$ and $\alpha$ are assumed to be constant and we consider an isopycnal depth in this part ($h'$).

Due to the fact that $Z$ and $h'$ (isopycnal line) in graph (Figure16 a, b) are from $L_1$ to $L_2$. we can calculate $h'$ and $Z$ values at ($x=L_1$) and ($x=L_2$), Substituting Eq. (4) in Eq. (3) and using these assumptions and that $v$ the geostrophic flow speed is assumed constant and is given by the slope of $h'$ in x direction, we have:

$$Q_V = \frac{g'}{f} \frac{H_1 - H_2}{L_2 + L_1} \left[ \left( -\frac{A}{\alpha}(e^{-\alpha L_2} - e^{\alpha L_1}) - \frac{a}{3}(L_2^3 + L_1^3) - \frac{b}{2}(L_2^2 - L_1^2) \right) \right] \qquad \textbf{(5)}$$

Where:

$$a = \frac{H_2 L_1 + H_1 L_2}{L_1(L_2^2 + L_1 L_2)}$$

$$b = \frac{H_2 L_1 + H_1 L_2}{L_2^2 + L_1 L_2} - \frac{H_1}{L_1}$$

If we assume that $L = \frac{|L_1| + |L_2|}{2}$, we have

$$Q_V = \frac{g'}{f} \frac{H_1 - H_2}{L_2 - L_1} \left[ \frac{2A}{\alpha} \sinh(\alpha L) - \frac{2}{3} aL^3 \right] \qquad \textbf{(6)}$$

Where:

$$a = \frac{H_2 + H_1}{2L^2}$$

$$b = \frac{H_2 - H_1}{2L}$$

$$A = \frac{H_1}{e^{\alpha L}}$$

$$\alpha = -\frac{1}{2L} Ln \frac{H_2}{H_1}$$

To obtain the Eq. 6, we defined $L$ based on $L_1$ and $L_2$ because $L_2 \neq L_1$. Although the minimum of $Z$ is not exactly at $x=0$, as it does not create large error. To show this in reality, the $Q_V$ is calculated separately with Eq. 5 and 6. The results show that the difference is about 2-5 percent, when using (Eq. 5) without any assumptions ($L \sim L_1 \sim L_2$). Another important point is the geostrophic velocity is considered under $R_0 \sim 0.1$ based on sections 2. For this reason, the Rossby number is calculated in the beginning of discussion on the main features of the deep flow because due to this assumption, the accuracy of methods (geostrophic law) would be evaluated.

To calculate the mean monthly volume flow rate of the deep current that enters the southern basin of the Caspian Sea, we assume that its density is greater than 1008.78 kg/m$^3$. Then the average density (for different seasons) of the flow below the upper boundary (e.g. Figure16) and use of equation (6) and figure16 are used for this purpose.

For the times that the middle and southern basins are filled, first the volumes of middle ($V_M$), and southern ($V_S$) basins (see Figure 16c) are calculated below three levels ($z=0$, $z=-100$, and $z=-150$ m, the approximate depth of the Absheron sill, which would be more appropriate only for the southern basin), then if we assume the same annual mean value of $Q_V$ for both basins, these filling times are estimated. The results of these calculations and comparisons between them for different seasons are

given in Tables 3 and 4.The results show that the maximum and minimum flow rates of abyssal water that enter the Southern Caspian Sea are in May and November respectively. In order to check the accuracy of Eq. 6, the $Q_V$ is also directly calculated from the numerical simulations without any assumptions. As shown in table 3, the numerical model value is greater than that of the analytical estimation. This underestimation by Eq. 6 can be due to the fact that we use some assumptions to obtain Eq. 6. The velocity used is the geostrophic velocity and the isopycnal line are also simplified. Apart from this, some errors come from the choice of 1008.78 kg/m$^3$ contour as its position changes for different months. If we consider the contour 1008.9 kg/m$^3$, it is appropriate for some months like January, but it is not useful for November because the maximum contour is 1008.8 kg/m$^3$ for this month. This method can create some error in some month, for example for January; the contour does not exactly reach the bottom. To solve this problem, we can use different contours for each month, however we follow one method for all months (under the same conditions for all months), as approximate estimates. The flushing time is then estimated based on the direct calculation from the numerical simulations. The results show that the flushing time is about 6-7(middle basin) and 13-14 (southern basin) years for the numerical simulations based on $z=0$.

As a result, the Eq. 6 can be useful to estimate the volume flux of water in the Strait, particularly in this oceanic environment without ADCP data. For example in Persian Gulf, there are many CTD data on the Hormuz Strait based on Bidokhti and Ezam (2009) but there is not any ADCP data at Absheron Strait, so the result of this paper can also be useful for estimation of the volume  flux  of overflow water in their work.

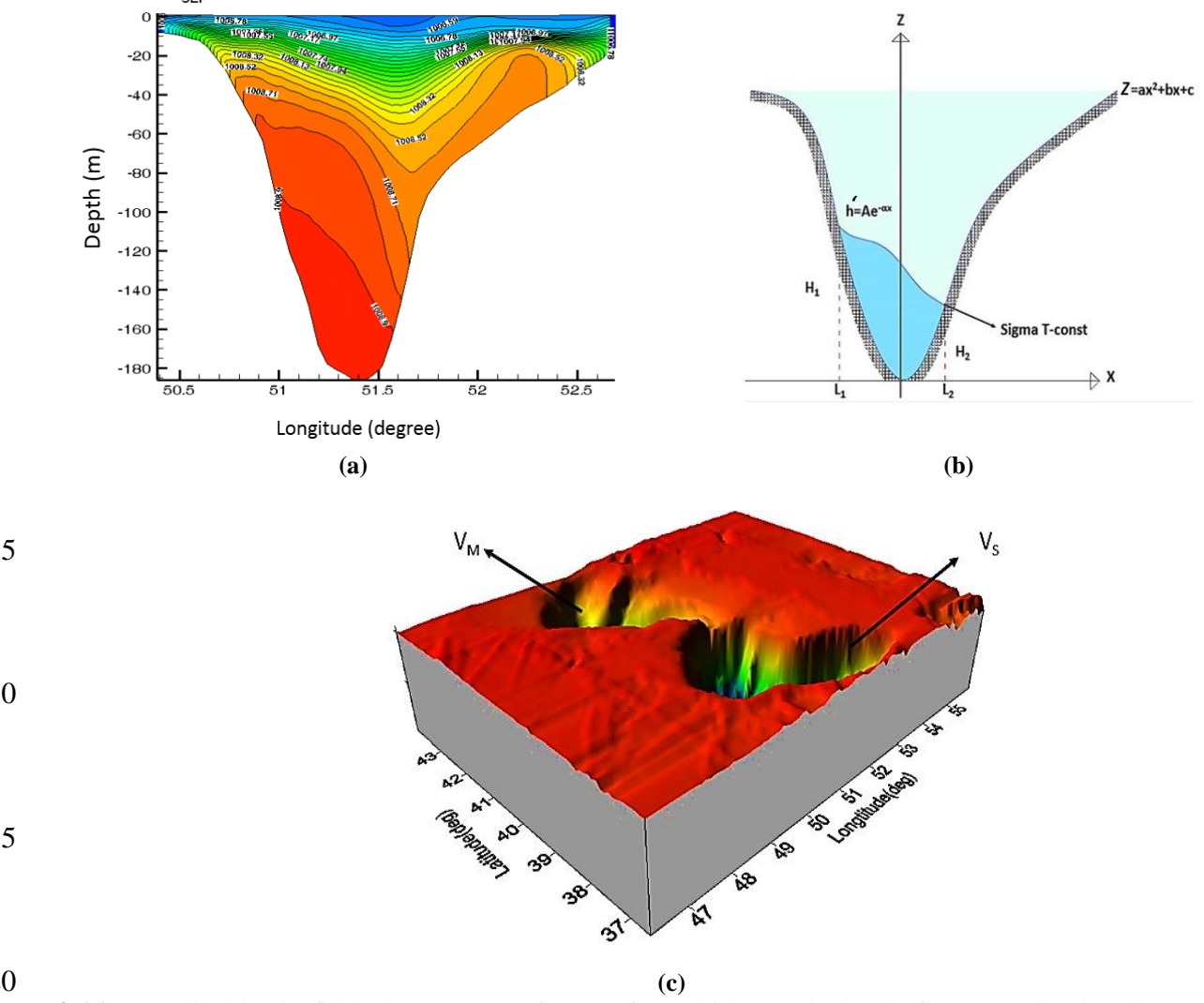

**Fig16**: (a) Typical density fields along transect I for Sept. from which we calculate the flow rates. (b) The scheme of the topography with a typical isopycnal and model parameters. (C)The model bathymetry used to calculate the volumes of middle ($V_M$), and southern ($V_S$) basins. The Surfer software are used to plot and calculate $V_M$ and $V_S$ by GEBCO data with 0.5° × 0.5° resolution.

**Table3:** The model boundary current parameters (1 $Sv=10^6$ m³/s) for different months.The last column (red symbol) show direct calculation from the numerical model

|  | $H_1$ (m) | $H_2$(m) | $2L$ (m) | $Q_V$(Sv) Analytical | $Q_V$(Sv) Numerical |
|---|---|---|---|---|---|
| **NOV** | 55 | 10 | 19000 | 0.016 | 0.034 |
| **JAN** | 145 | 85 | 32000 | 0.115 | 0.15 |
| **MAY** | 145 | 55 | 34000 | 0.146 | 0.17 |
| **SEP** | 135 | 45 | 27500 | 0.116 | 0.16 |

**Table 4**: Flushing times of the middle($T_M$) and the southern ($T_S$) basins (using an annual average volume flow rate ($Q_V$) below three levels based on formula 6.

| Level | $V_M$(m³×10¹³) | $V_S$(m³×10¹³) | $T_M$(year) | $T_S$(year) |
|---|---|---|---|---|
| z=0 (sea surface) | 2.55 | 5.12 | 8.35 | 16.77 |
| z=-100 | 1.09 | 4.13 | 3.57 | 13.5 |
| z=-150 | 0.36 | 3.62 | 1.17 | 11.85 |

## 4.2 The importance of deep flow in the southern basin

In this section, the importance of deep flow is discussed. In general, deep flows play pivotal roles in ventilation of deep part of the water in the Caspian Sea. We can observe the sign of life in the deep part of the Caspian Sea, especially in the Southern basin (Terziev, 1992), however the reasons for this have not been addressed clearly so far. This is the main reason why the deeper part of the Sothern basin is not "dead" like that in the Black Sea. This flow carried oxygen from the surface to the bottom layers, and nutrients from the bottom towards the top by a slow advection. As a result of this, the overflow can be considered the most important element of this ecosystem. However, if we look at the other issues of this enclosed sea, we find other interesting and determining points. These days, Oil well pollution and climate change effects are the most important problems in the Caspian Sea. To begin a discussion, it is interesting to look at the locations of the oil wells in this Sea. As the location of the oil and gas wells in this basin are shown in figure 17a, it can be seen that they are mainly situated on two areas in the northern basin and particularly around Apsheron sill. Also a satellite image shows that some oil spills in have occurred in the vicinity of these oil and gas wells around the sill. For example, the figure 17b which is extracted from Marina and Yu Lavrova (2015) using satellite data, shows that the spills are located on the sill and also in the western parts of the southern basin. In addition to this, studies on the sea bed on the Apsheron strait and Bako Gulf indicated that approximately 1 and 1.5 meters covered with sludge and oil residues in the form of high-density pellets and mazut (Escani and Amini, 2013). If we consider all of the points and look at the path of the deep flow (Fig.8), the present work can be very important for consideration of the impacts of such oil exploration activities and the fate of deeper as well as other depths of this environment, if certain careful actions are not taken.

By considering these points and a glance over the whole this research, it is concluded that we always pursued a goal. In section 2, after showing our evidence about the existence of deep flow, it is focused on the sinking and mixing processes, as such flow is very important in the ventilation of southern basin. Unfortunately, the pollution of oils can spread into deeper parts of the southern basin. In other words, this flow, although plays a positive role naturally, but with such human activities it can have a negative role as well in this area, as well as in the oceanic thermocline circulations if world-wide oil exploration should take place in certain high latitudes.

$\lambda$, the overflow direction was calculated showing that the flow passes over some wells area in Apsheron sill. Due to this angle, the flow passes through the wells near the Azerbaijani Republic rather than the eastern part of the sill near Turkmanestan. In

section 3, the dynamics of the flow was discussed. Among all of the dynamics aspects, it was found that under certain conditions the flow was separated from the cape due to the fact that two eddies were formed here. Eddies are very substantial in the ocean and climate (Gill, 1984) because they advects mass (here the oil pollution) and their ability to propagate is crucial to its contribution to mixing rates (Friel, 1987). Based on this result, it can be concluded that the region near the SefidRud

5   Delta cape may be the most polluted in deeper parts (present work) and surface (Fig.17b).

In section 4, it is tried to estimate the flushing time because it is very important for the ventilation of the Caspian Sea basins. In addition to this, in section 2, it was discussed why this time is important for the required time for running a numerical model, particularly for deep part proper adjustment.

 When it comes to climate change, this problem leads to many problems in this area like rising the sea level of water (Chen et

10   al., 2017). The main reason for this maybe an upward trend of temperature in recent years. Due to this, the question is raised in our mind as what is the effect of climate change on the deeper flows of the Caspian Sea. Based on the section 2, it is mentioned that the water sinks in the northern basin and after filling the middle basin; finally, it overflows into the southern basin. If it is accepted that the atmospheric temperature rises it will warm in the northern part of Caspian Sea, and it is predicted that the sinking process will be weaker. As a result of this, the volume of water mass by the overflow will decrease and flushing

15   time would increase. In other words, it probably has a negative effect on the deep part of the southern basin due to weaker ventilation by this flow, if it does not totally shut off this ventilation.

The main goal of this paper was to investigate the deep flows in the Caspian Sea, and also it was attempted to consider some of their environmental aspects and effects while discussing the importance of such flows for the future of the Caspian Sea environment.

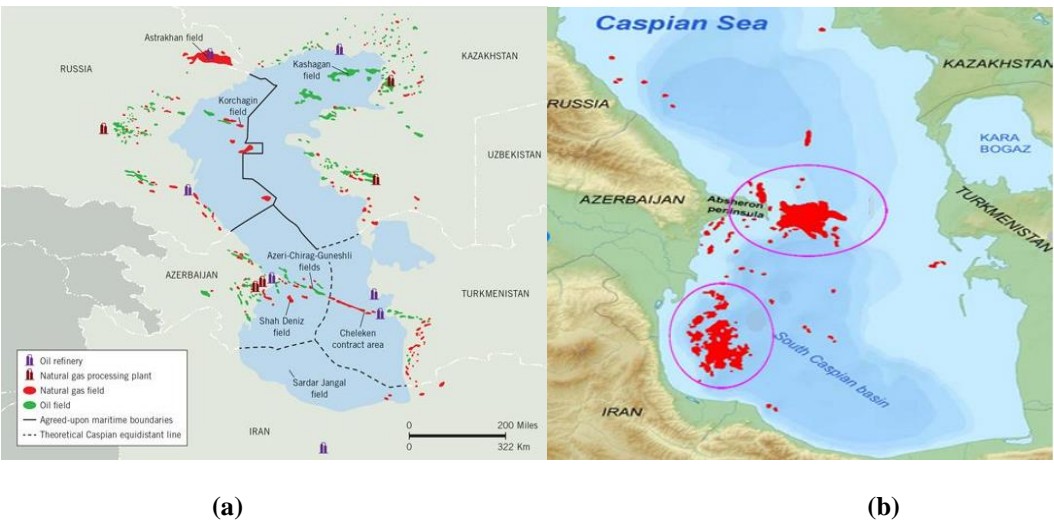

(a)                                                                                   (b)

**Fig 17**: (a) The location of oil and gas fields in the Caspian Sea, which is extracted from *https://www.offshore-mag.com*. (b)The map of oil spills revealed from satellite radar imagery in the central and south western parts of Caspian Sea in 2010. (Mityagina and Yu Lavrova, 2015).

## 4. Conclusions and consequences

The results of observations and numerical simulations showed that there is an abyssal flow from the middle to the southern basins of the Caspian Sea. The density difference between the deeper water of the middle basin and that of southern basin leads to an overflow gravity current over the Absheron sill. This difference is mainly due to the temperature difference between deeper parts of these two basins, as a result of cold water initially sinking in the northern part of this Sea, at about 48 degrees latitude, that fills first the middle basin and then overflows towards the southern basin. In the autumn and winter, surface water cools and its density is increased and then it sinks to the deeper parts of the middle basin, like deep convection process in high latitude oceans. Winter storms and cold wind provide the cooling of this rather high-latitude shallow water in the northern basin.

We also estimated typical mass transport and flushing times of the deep-water basins of this Sea. After the sill, the flow adjusts itself moving south as a gravity driven topographically trapped current, spiralling into deeper parts due to bottom friction and entrainment. It always tends to move toward the western shores of this Sea, mainly due to the Coriolis force that shifts it to the right. Such flow is important in the abyssal circulation and ventilation of the deep southern basin of the Caspian Sea. For vorticity and potential vorticity of the flow, the formulas which are presented by Falcini and Salusti (2015) are used to estimate the changes of relative vorticity and potential vorticity over Absheron sill in the trapped current.

Results also showed that nearly $3.05\times10^{12}$ m$^3$ of water per year by this abyssal flow can enter the Southern basin, giving a typical flushing time of about 15 to 20 years which are of the same order as those estimated by Peeters et al. (2000). Some points are discussed as how the southern Caspian Sea basin ecosystem can be strongly dependent on this flow.

The northern and middle Caspian Sea basins have become important areas for oil and gas explorations (especially the Absheron shallow Strait area) and marine transport nowadays. Since the Caspian Sea is an enclosed sea, the adverse effects of such activities may particularly affect the deeper parts of the Caspian Sea basins. For this reason, it is recommended that more detailed observational data are collected in the deep parts of the southern and middle basins of the Caspian Sea, by joint projects with neighbouring countries. More extensive and fine-resolution observational data and numerical simulations are required to find more details of the overflow structure over and around the Absheron sill (Strait) and the deeper parts of this Sea basins.

**Acknowledgment**

The financial support of University of Tehran, while doing this work is greatly acknowledged. Numerous comments of Prof. J. M. Huthnance in improving the paper is greatly acknowledged.

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
