# Peer review of "Some aspects of the deep abyssal overflow between the middle and southern basins of the Caspian Sea"

_Ocean Science, 2018_

## Referee Comment (RC1) · C. W. Hughes (Referee) · 11 May 2018

This paper investigates the overflow of dense water from the middle to the southern basins of the Caspian Sea, using an ocean model and a variety of analytical models, together with some in situ measurements. It begins and ends with a good description of the background geography and dynamics, which lead to the seasonal overflow between the basins.

The main conclusion given is the flushing time for the southern basin, which is estimated at between about 12 and 17 years.

[Figure]

Although there are some interesting results and ideas here, the presentation is extremely disordered and poorly evidenced. Although the authors have a model run at their disposal, they do not use the model results other than as qualitative pointers to parameters to use in the analytical models. Furthermore, the analytical models are not tested against the model results to see whether they are appropriate. Finally, the three analytical models are not clearly differentiated to explain how/where they are expected to apply, and terminology is poorly defined. In one case, this leads to a contradiction in which h is simultaneously used for the height of the interface (to determine the geostrophic current) and the thickness of the bottom layer (in the definition of PV) over a sloping bottom topography.

As a result of these problems I feel that a lot of work needs to be done before the paper meets the standards expected for Ocean Science.

The main result given, concerning flushing times, uses the model stratification to infer a flow (volume transport) over the Absheron sill, via a 1.5 layer model and some analytical approximations to the topography and interface height based on the model results. This volume transport could have been diagnosed directly from the model, with no approximations, and giving a time series as a function of interface density. Only then does it make sense to use the 1.5 layer model to test whether such a method produces a good estimate of the transport. If it does, then observational data could be used in the same way.

The other analytical models are presented with no real reason given for using them. Are they supposed to predict the path and strength of the overflow current? If so, they should be compared with the model output to see whether the prediction is good.

More generally, some bits of observational data are shown, and some bits of model data, but with virtually no comparison apart from one surface current record. It is not even clear from the description how much observational data is available: are the CTD data all one-time measurements? How many casts make up the (almost unreadable)

plots in Figure 4? Equally, many of the plots seem to have randomly chosen axis labels or contour intervals which make them very difficult to interpret. In particular, Fig. 19 has many contours with tiny labels, but none match the reference contour used in the overflow transport calculation (which does not even intersect the seafloor in the January case). In a similar vein, model diagnostics are discussed on 10 different sections, A-E and I-V, with some subset of these actually shown. There is no reason given for having two different sets, and many different map regions and aspect ratios are used (in Fig. 9 the region is split into two shown side by side, for no clear reason).

I could see a route to making a good paper along the lines of the following:

1) Show observations along with comparable data from the model (the same variable and the same plot style!) to validate use of the model.

2) Diagnose the overflow from the model, calculate and discuss flushing periods.

3) Use only information available from observations to calculate an approximation to the overflow, and use the same approach in the model to validate it.

4) Investigate the flow downstream of the overflow in the model - show sections, derive parameters and compare to the analytical model.

A few particular issues:

P3 Lines 15-16 - ranges cited are not consistent with those shown in Fig. 1 - needs a comment.

Fig. 1: Why not label -5,0,5,10...deg C instead of -8... ? And the first of every other month rather than every 65 days as seem to be labelled here?

Fig. 2 (and later) - Would be better to use either sigma-theta or density minus some reference function of z. Sigma-T is not a very meaningful variable in deep water.

P4 Lines 6-7 - "there are contour gradients in the southern basin" nothing is shown from the southern basin, what is the basis for this statement?

Figure 4 - background colour is unhelpful and unnecessary. It would be helful to know the context: what is the depth here? How far from the coast are the observations? Where are the CTD casts that these contour plots are constructed from? It looks as if there must be more than the 3 marked on figure 2.

Page 6 - References should be given for the various data sources. Line 5 - what is the source of the January data used to initialise the model? Is this different from the few CTDs in figure 2? How is the initial temperature and salinity chosen where there are no observations?

Page 7 line 6 - "bottom left" should be west.

Figure 7 - needs to show the same variable (unclear if this is the case) and with the same contours for comparison.

Fig. 8 - should have the same colour scale for both, and the zero contour should be made clear.

Fig. 9 - Maps should not be split across two panels.

Page 10 - The assumptions underlying the model should be set out more clearly, and its origin (reference or derivation). It is essentially that for a freely-sliding particle on a slope under reduced gravity, but could be interpreted as a streamtube with appropriate background. "Assuming no pressure gradient" is not consistent with the earlier statement that it is geostrophic, but the g'tak(theta) term actually represents a pressure gradient term. You should also make clear that x is directed along the slope and y perpendicular to it, rather than east and north. "H is the depth of the overflow" is ambiguous - the depth of the sill? the depth of the interface between dense and less-dense water? The thickness of the overflowing layer?

Page 12 - the later discussion of E values should be included here, otherwise it is too hard to follow. What value of Ri is actually used? Saying Ri > 0.08 is not very helpful. Similarly, given the formula for r_b, a reason for choosing 2x10ˆ-5 should be given.

[Figure]

Page 13 - This really needs a systematic comparison with the model results - more than just saying that the model shows the flow sinks by 180-200 m (i.e. to the bottom).

Page 15-16 - the xi, psi coordinate system is described, but never used except in the axis label on Fig. 15. You seem to still be using x, y. There is a missing ")" in (5). Formatting of the equations is very strance and hard to read. Also, what assumptions go into this model? Is PV assumed constant across the flow? How can that be possible at the boundaries where layer thickness tends to zero? What input values were used to calculate the results in Figure 15? Is there any evidence that any of this is realistic? You have the actual values in the numerical model, does the analytical model predict these with any skill?

Page 17 - in (7), h is an interface height (or depth?) relative to a fixed level, whereas in (9) it is a layer thickness. These are incompatible if the flow is over topography as described.

Page 18 - line 3 "after the sill the flow depth is not changing and the entrainment effect is almost zero" - what is the justification for this statement?

Page 18 - line 5-6 "V is the downstream of the sill and D is the location at which the current is trapped by the topography" - I can't understand what this is trying to say.

Page 18 - lines 9-12 - why should the flow vanish at this boundary and not the other? This guarantees that h (whatever it is) must increase away from the boundary at which the condition is applied.

Line 15 - needs to explain that 1/beta is considered the relevant Rossby radius for this problem.

Page 19, lines 36-37 - you state that vorticity is predicted correctly and that the model confirms a prediction, but present no evidence for this.

Page 20 bottom line - I think you are also assuming that |L1|=|L2|.

Page 21 - why assume the outflow from the middle basin to be the relevant flow to calculate the flushing time? Surely the inflow is more relevant. What is the meaning of h in Figure 18?

Figure 19 - contour labels are virtually unreadable. The contour interval is not anything obvious, and the critical contour used for calculations (1008.78) should be highlighted (it is not even one of the plotted contours). In fact this contour doesn't reach the bottom anywhere in the January plot, making it impossible to identify meaningful parameters in this case.

Table 5 - I can't find a way to get values from Fig. 19 that agree with 2L values in this table. Are these actually L?

---

## Referee Comment (RC2) · Anonymous Referee #2 · 31 May 2018

This manuscript discusses overflow of dense water from the Middle to the South Caspian basin. There are some previous hydrographic observations but the study is mainly via models with some guidance by the observations. A 3-D numerical model is run for several years' simulation. Analytic models, with somewhat simplified dynamics and guided by the numerical model output, are used to discuss behaviour of the flow downslope from the sill in relation to (i) amount of friction and entrainment (ii) vorticity evolution (iii) depth as a function of cross-flow coordinate, and (iv) corresponding transport and flushing times for the two basins.

These are interesting topics, although specific to the Caspian as discussed here. There

ought to be some discussion about the circumstances in which the simplified / analytic models and approaches here would be applicable and useful in other contexts; this would make the manuscript interesting for more readers. A related concern is that there should be better motivation and synthesis of the different forms of evidence. The numerical model is compared with some rather irrelevant current observations, not very convincingly, and more favourably but only very briefly with relevant hydrography. Curiously, the numerical model is used to provide values of variables or parameters to the idealised analytic models which are used to discuss dependencies ((i) to (iv) above). One might have expected that the numerical model itself could be used; presumably it has all the dynamics (except perhaps entrainment) of the simpler models. Moreover, one might expect more comparison between the results of the numerical and simpler models. Obviously the simpler models are easier to use for exploring dependencies, but are they good enough?

The manuscript is structured reasonably and the English is generally understandable. The following "Detailed Comments" are for minor revisions but I think the issues above warrant more substantial revisions.

Page 2 Lines 2-18 are not made to relate to the Caspian. I think an alteration might be made at line 13, e.g. ". . but also in ventilation of semi-closed and closed basins, e.g. the Caspian Sea. Study . ." Lines 17-18 seem misplaced; they are not made to relate to the previous or following text.

Page 3 Line 16. "7-10" does not quite correspond with figure 1. Line 19. "16" does not correspond with lines 15-16 or figure 1.

Figure 2. Both sides would benefit from a distinct coastline. The left side should have the same latitude and longitude scales.

Page 6. Line 3. "layers" not "levels". Line 14. I am not convinced by "are rather consistent with observations". However, currents here are not very relevant to the sill and overflow there. Line 16 "as can be expected" and line 17 "interpolation". The

uncertainties due to the model grid might be estimated by comparing the variance of (model – observation) with the variance of (difference between adjacent model grid points). Figures 7 and 8 are probably better evidence that the model is working OK for the purpose of this study.

Page 7. Lines 4-5. I think this sentence "This . . Absheron." does not add information. Lines 5-6. I think "From . . derived." belongs at the beginning of section 3. Line 7. Refer to figure 9 as well.

Page 8 Lines 37-41. I think this belongs in section 3 before section 3.1.

Page 10. Line 5. "No pressure gradient". There are horizontal gradients of density implying horizontal gradients of pressure. However, I think equations (1) can be OK if understood as in coordinates parallel to the slope. Equations (1). I think there might be some comparison with the model of Shapiro and Hill (1997) J Physical Oceanography, 27(11), 2381. It is very similar albeit steady-state. Line 19. In section 3.1 "re" and "rb" only appear summed as "rb+re" and a symbol for "rb+re" would be useful.

Page 13. Table 1 columns could be fitted to contents so that row 4 is all on one line. Line 14. "direction" not "horizontal". Line 18. Word order better ". . flow is trapped after about 10 km. . ."

Figure 12. The x and y scales in the left panel differ. Ideally they should be the same but if not the caption should say that they differ.

Page 15. Lines 4-10 should somewhere state the assumption of steady flow. Equations (5), (6). In the integral exponents (of e) I think the integrand should be r/u where u has an overbar. "r" needs definition.

Figure 15. In the left panel, left axis, the variable should be $\zeta$ Page 16 line 39 to page 17 line 1. ". . the graph shows decreases from I to IV . ."

Page 17 Line 5. Omit first "sea". Line 6. "similar assumptions" – but also steady which should be stated. Line 8. "No mixing could exist" - not true; no mixing is an assumption.

[Figure]

Line 10. "0.00002 s**-1" should be related to 0.003|U|/H and values of U, H.

Page 18 Line 4. "short distance between D and V" should be shown on the same figure. How are locations I, . . V defined? Lines 5-6. "D is the location at which the current is trapped by the topography". What about C and E for example? Line 15. To obtain R from (11) involves a value for potential vorticity $\Pi$. How is this estimated?

Page 19 line 36. "The numerical model . . confirms this prediction." The reader cannot infer $\Pi$ quantitatively from figure 9.

Page 20. Equation (13). This form for h differs from (10). Equation (14). ")" missing after exponentials. Last line. "If we assume . ." does not give (15) directly: L2 $\neq$ L1. Need to say ". . we approximate (14) by". Then it makes sense to compare the values given for Qv by (14) and (15) (page 21 line 5).

Page 21 lines 10-11. Better to move "are calculated" to before "below"

---

## Author Comment (AC1) · 25 Jun 2018

Many thanks to *Dr. C. W. Hughes* for comments on "Some aspects of the deep abyssal overflow between the middle and southern basins of the Caspian Sea" by JavadBabagoliMatikolaei et al Our answers and further clarifications are as follows:

**Answer to Dr. C. W. Hughes**

In the main paper, due to text limitation, we could not mention all our reasons for publishing this paper. Here are further reasons for presenting and using the methods in this paper:
*Dr. C. W. Hughes correctly mentioned that the most important result of this paper is the flushing time*. Why it is very important for us and of work?
In terms of importance, as mentioned in the paper, not much work on the deep flows and water circulations of the Caspian Sea basins, the largest water body in the world, have been done, hence, the present work may be the first attempt to concentrate on such issues which are crucial for the future fate of this sea, that have been partly addressed in the papers conclusion section.
Apart from this, there are two important points concerning the importance of this research. The first point is that there are a number of evidence which shows signs of life in the deep part of the Caspian Sea, especially in the Southern basin of Caspian Sea. For example, these signs in deep parts of this sea have presented in: Biological Features and Resources Caspian Sea, M.G. Karpinsky · D.N. Katunin · V. B. Goryunova · T.A. Shiganova, In Caspian Sea Environment, Springer, Part P (2005): 191–210).
This means that the Caspian Sea is not as Black sea which is due to lack of such ventilation is nonproductive in deep parts. But you cannot also find any research to answer this question as why the deeper part of the Caspian Sea is rather alive. The current study shows that such abyssal flow can ventilate the deeper parts of the Caspian Sea.
Another point is oil pollution due to oil exploration, especially in the Strait of Apsheron, which is underway that can affect the deep parts of the southern basin of this Sea (see Fig 1).

[Figure]

Fig 1. The location of oil wells in Apsheron strait (left) and the result of the model (right). The deep water exactly moves over the extracting points that can contaminate the deeper part.

As a result, deep flows and flushing time estimations are very important issues for evaluating the future of the Caspian Sea.

About the methods chosen in the paper, as we first began to investigate deeper flows based on existing observation data, they showed differences between the middle and southern basins in terms of density. In the first step, we tried to use analytic models with observation data. However, we found that there are not enough data to study the structure of the flow using the observation data only. We agree that the observation data for such research have no substitutions. For example, the most important points discussed are on the flow in the Strait of Apsheron and also southern basin. In these locations, the distance between the measured data is much larger than the expected current width, so we only see part of the flow and could not fully detect the current. Hence, we have to use partly some numerical model results to test analytical models.

Regarding the analytical model validation, we presented 3 analytics models in this study (Eq. 4, 13, 17). When it comes to Eq. 4, one of the best ways to test a model is to know how much water is sinking. We calculate this value 180-200 m based on Figure 11 and 18 (in the model) or Peeters et al (2000) figure 7. However, we tried to present further explanations of our approach. To deal with this, we chose transect A and C (in the paper) in September and used the density 1008.9 kg/m$^3$for calculations. The method of our calculation is shows fig 2.

[Figure]

$$\Delta H = H_2 - H_1 = -180 \text{ m}$$

Fig 2. The calculation of sinking based on the isopycnal lines, (left) over Absheron and (right) downstream of the Absheron sill.

In another (better) method, it can be calculated by Peeters et al (2000) measurements in figure 7 in the paper (fig. 3).

[Figure]

Fig 3. The calculation of sinking based on Peeters et al (2000) measurements.

Hence, the analytical model is well-matched with the results of field observations and the numerical modeling simulations. Although these figures are in the paper, in the new edition of the paper some explanations was added to the text to clarify the method.

When it comes to Eq. 13, we combine two methods to calculate the Rossby radius of deformation. We combine the method of Bidokhti and Ezam, 2009, and Falcini and Salusti, 2015 with some minor changes. Fortunately, both of paper published in Ocean Science and have also compared their model results with observation data. Due to your comment, we have added some comments on models results in the new version. For example table 1, shows the comparison between analytical model and numerical mode results. In table 1, the last column shows the direct calculation from the numerical model. Figure 4 shows our method to calculate $R$ from the model.

Table 1: Boundary trapped current model parameters and values of $R$. The last column shows the result of the numerical model

|  | $h_0$(m) | $h_1$(m) | $h_R$ (m) | $R$ (km) Analytical | $R$ (km) Numerical |
|---|---|---|---|---|---|
| NOV | 120 | 270 | 400 | 17 | 18 |
| JAN | 150 | 440 | 485 | 15 | 16 |
| MAY | 110 | 300 | 400 | 30 | 34 |
| SEP | 140 | 430 | 570 | 25 | 35 |

[Figure]

Fig. 4: The method of calculation of Rossby radius of deformation from the numerical model in the deeper part of the southern basin.

About Eq.17. This comment is very interesting and we have added some clarifications and the flow rate from the numerical results to the new version of paper. For example, you can see the comparison in table 2.

Table 2: The model boundary current parameters (1 $Sv=10^6$ m³/s) for different months. The last column shows direct calculation from the numerical model results

|  | $H_1$ (m) | $H_2$(m) | $2L$ (m) | $Q_V$(Sv) Analytical | $Q_V$(Sv) Numerical |
|---|---|---|---|---|---|
| NOV | 55 | 10 | 19000 | 0.016 | 0.034 |
| JAN | 145 | 85 | 32000 | 0.115 | 0.15 |
| MAY | 145 | 55 | 34000 | 0.146 | 0.17 |
| SEP | 135 | 45 | 27500 | 0.116 | 0.16 |

The reasons for using three analytical models are that the dynamics of the current changes when moving along the strait and then trapped along the bottom slope in the southern basin, so we used three analytical models to describe the dynamics of different stages the flow. To explain more about the presentation of three models, as in the Iranian seas (Strait of Hormuz and Apsheron sill) there isn't any ADCP data to understand some important parameters of outflow in basins, we used these models. For example, in the Hormuz Strait, CTD data could be the only data to specify the outflow from the Persian Gulf that are non-extant appropriately. Hence, thanks to this kind of model that can used to calculate many parameters of outflow without using ADCP data, for example, the volume of outflow due to using the Eq. 15.

About presentation, we agree with your opinion when it comes to using some parameter such as $h$ in different meanings. We have faced with some problems to present the model because we have to define many parameters due to using three analytical models. In the new version, we have changed some parameters and added some explanation about our methods.

About observation data, we used two kinds of observation data, CTD and ADCP, in this paper. The CTD data are obtained from UNESCO Atomic Energy International Agency, for summer

1996. This data is recorded from 25 August to 9 September. The data are collected at 27 stations using an Investigative ship in this project namely Hajef. ADCP recorded by RCM9 current meters (at the ADCP station) at 3 depths on a mooring, near the surface, 50 m, and 200 m near the Iranian coast. We used this data to validate the numerical model. This data is the only exiting one that are collected in the southern Caspian Sea. For the method of these data collection, please see:

Ghaffari, P. &Chegini, V. (2010). Acoustic Doppler Current Profiler observations in the southern Caspian Sea: shelf currents and flow field off Feridoonkenar Bay, Iran. *Ocean Science*, *6*(3), 737.

**Other comments of *Dr. C. W. Hughes**

1) Show observations along with comparable data from the model (the same variable and the same plot style!) to validate the use of the model.

Although we accept your suggestion, according to what was mentioned, observational data are very limited. In some places, we compared exactly the observation data with numerical model, for instance the figure 5 and 7. We confirm that if there were sufficient data, there would be no reason to use the numerical model. We emphasize that only observation data (CTD) is used to find the existence of the gravity driven flow between the middle and the southern Caspian, as Peeters et al (2000) measurements also indicate.

2) Diagnose the overflow from the model, calculate and discuss flushing periods.
In the new version, the flushing time is calculated directly from the numerical model as well. The calculation shows that the flushing time is about 6-7(middle basin) and 13-14 (southern basin) years for direct calculation from the numerical model results based on z=0.

3) Use only information available from observations to calculate an approximation to the overflow, and use the same approach in the model to validate it.
Above discussion may be enough for this question.

4) Investigate the flow downstream of the overflow in the model - show sections, derive Parameters and compare to the analytical model.
We add this part in the new version of the paper. We compared the numerical model results with the analytical model ones based on your interesting comments.

A few particular issues:
P3 Lines 15-16 - ranges cited are not consistent with those shown in Fig. 1 - needs a Comment.

The difference comes from the years that have been chosen in the research and are not the same as fot Ibrayev et al., 2010, and also, we used data every 6 hours, but probably they used monthly average data. Some more sentences are added to the paper for clarification.

Fig. 1: Why not label -5,0,5,10...deg C instead of -8... ? And the first of every other month rather than every 65 days as seem to be labelled here?

There is not any special reason for temperature axes. The output itself is in the Excel format and the changes are done in the new version. Regarding the months, if every month is given, the axis will be crowded and could not be readable. In this axis, since the series has been for two years, the year should also be given.

Fig. 2 (and later) - Would be better to use either sigma-theta or density minus some reference function of z. Sigma-T is not a very meaningful variable in deep water.

We accept your comment although density should only be used in figure 3 due to a depth of basin. In figure 4, due to the depth of water on the Strait (180-200 m), we can use Sigma-T. In another part of the paper, the density is used for the calculation. Hence figure 3 has been changed in new version.

P4 Lines 6-7 - "there are contour gradients in the southern basin" nothing is shown from the southern basin, what is the basis for this statement.

It means that figure 4 shows the outflow due to the slope in the sigma-T contour, and figure 3 expresses the density differences between the middle and southern Caspian Sea basins. However, your comment is right and in the new version, it is corrected because our observation data are for the Absheron strait.

Figure 4 - background color is unhelpful and unnecessary. It would be helpful to know the context: what is the depth here? How far from the coast are the observations? Where are the CTD casts that these contour plots are constructed from? It looks as if there must be more than the 3 marked on figure 2.

Regarding your opinion, we redefined figure 4. But we think that the figure in the paper is better than the new figure. This is not a specific problem and can be replaced in the final version. The depth of water in this area is about 180 to 200 meters. The distance of Azerbaijan coast is about 80 km from this area. In figure 2, we have identified the geographic location of observational data (see transect B, Fig.2). This information is added to the text in the new version.

[Figure]

Fig. 4: different background color for Fig.4 in the paper.

Page 6 - References should be given for the various data sources. Line 5 - what is the source of the January data used to initialize the model? Is this different from the few CTDs in figure 2? How is the initial temperature and salinity chosen where there are no observations?

Yes. It is corrected. It means that the model was initialized for winter (January) using monthly mean temperature and salinity climatology obtained from Kara et al (2010).

Page 7 line 6 - "bottom left" should be west.

Corrected

Figure 7 - needs to show the same variable (unclear if this is the case) and with the same contours for comparison.

Although your suggestion is quite good, we cannot compare them because the numerical model (present work) and Peeters et al (2000) are not for the same years. Comparison is meaningful when the two quantities are the same. We emphasized this in the revised paper in that the overflow occurs in the Caspian Sea on the Absheron sill. However, we tried to apply your opinion in the new version.

Fig. 8 - should have the same color scale for both, and the zero contour should be made clear.

Corrected

Fig. 9 - Maps should not be split across two panels.

We tried this method but the details of the current are not displayed. Typically, according to your comment, in Figure 5 we have a sample.

[Figure]

Fig. 5: another style of display Fig.9 (in the paper).

Page 10 - The assumptions underlying the model should be set out more clearly, and its origin (reference or derivation). It is essential that for a freely-sliding particle on a slope under reduced gravity, but could be interpreted as a stream tube with appropriate background. "Assuming no pressure gradient" is not consistent with the earlier statement that it is geostrophic, but the g'tak(theta) term actually represents a pressure gradient term. You should also make clear that x is directed along the slope and y perpendicular to it, rather than east and north. "H is the depth of the overflow" is ambiguous - the depth of the sill? the depth of the interface between dense and less-dense water? The thickness of the overflowing layer?

We used the streamtube model in the section 3-2. In this part, the path of the flow is more important for us. We are going to examine the path of particle motion based on the Lagrangian view. We wrote the momentum equation for particles and the most important assumption is that we considered entrainment effect as well as friction. Eq. 1 is right because we consider the coordinates parallel to slope.
 $H$ is the thickness of the overflowing layer.
Origin of the overflow is from the top of the sill between the two basins. About stream tube, we use this method in section 3.2 to study of vorticity when the flow moves over the strait. More clarifications are also given in the new version.

Page 12 - the later discussion of E values should be included here, otherwise, it is too hard to follow. What value of Ri is actually used? Saying Ri> 0.08 is not very helpful. Similarly, given the formula for r_b, a reason for choosing 2x10^-5 should be given.

To calculate Richardson, we use $Ri = \frac{g'H}{U^2}cos\theta$. Based on our result, U which is 0.1 to 0.2 m/s, g'=0.00222-0.00251m/s$^2$,H=50-70 m, tanθ=0.02. $Ri = \frac{0.00251*50}{0.2\times0.2}0.99$~ 3.1. Based on Ashida and Egashira (1997)E*=0.0015Ri$^{-1}$. Hence, E*~0.0005 and re~E/H=0.0005/50= 1×10$^{-5}$.

According to the Cheng et al, 1999, $C_d \sim 2\text{-}6 \times 10^{-3}$ the drag coefficient is not a constant value. Hence, we consider two value for $C_d \sim 3 \times 10^{-3}$ and $5 \times 10^{-3}$. As a result of $r_b = C_d u/H = 0.005 * 0.2/50 = 2 \times 10^{-5}$. This calculation was added to the main text.

Page 13 - This really needs a systematic comparison with the model results - more than just saying that the model shows the flow sinks by 180-200 m (i.e. to the bottom). This was discussed above. Based on your opinion, some extra explanation was added to the new version of the paper.

Page 15-16 - the xi, psi coordinate system is described but never used except in the axis label on Fig. 15. You seem to still be using x, y. There is a missing ")" in (5). Formatting of the equations is very strance and hard to read. Also, what assumptions go into this model? Is PV assumed constant across the flow? How can that be possible at the boundaries where layer thickness tends to zero? What input values wereused to calculate the results in Figure 15? Is there any evidence that any of this is realistic? You have the actual values in the numerical model, does the analytical model predict these with any skill?

Falcini and Salusti (2015) presented an analytic model for vorticity and PV ($\xi$, $\psi$) coordinate system. We use this system to estimate the relative vorticity when moving on the sill. After the sill, dynamics of the flow change and appear as a trapped flow. Hence, in this paper, we used ($\xi$, $\psi$) coordinate system only to calculate relative vorticity based on Falcini and Salusti method. Potential vorticity is assumed constant along the flow until the slope of the isopycnal is zero, hence we consider this assumption. Figure 15 is plotted based on the Eq. 5. The input values are some quantities such as u, $\partial h/\partial x$…. these quantities are extracted from the numerical model results.The first step is an estimation of $\Pi$ moving on the Apsheron sill. On the other hand, Bidokhti and Ezam considerd that the potential vorticity was conserved, when moving from the sill, as they ignore friction and entrainment. They presented their analytical model to calculate Rossby radius of deformation when the flow was trapped on the topography. We try to develop this theory by considering the bottom friction and entrainment. The problem comes from this reality that the potential vorticity is not conserved based on our assumption. As a result, we estimated the changes of vorticity on the strait based on Eq.5 from Falcini and Salusti (2015). In terms of its accuracy, we add this part to the new version and also discussed above.

Page 17 - in (7), h is an interface height (or depth?) relative to a fixed level, whereas in (9) it is a layer thickness. These are incompatible if the flow is over topography as described.

In this part, we tried to use the previous paper published in the ocean science (Bidokhti and Ezam, 2008; Falcini and Salusti, 2015). In this model we define two layers flow together with boundary conditions. This method is better than using the bottom as a reference because the depth from bottom is not equal along the isopycnal.
In this model the two layers move together and we can define boundary conditions (BC) better when considering the surface BC.
In Eq. 7 and 9, the definition of h are the same in both of Eqs.

Page 18 - line 3 "after the sill the flow depth is not changing and the entrainment effect is almost zero" - what is the justification for this statement?

This indicates that stretching term is not important in vorticity because for this kind of flow the depth changes that can change vorticity. The entrainment effect is almost zero because Ri~ 50 and based on table 1, this effect can be ignored.

Page 18 - line 5-6 "V is the downstream of the sill and D is the location at which the current is trapped by the topography" - I can't understand what this is trying to say.
Yes. It is not clear and we added more clarifications in the new version so the reader can understand the location of V and D simultaneously.

Page 18 - lines 9-12 - why should the flow vanish at this boundary and not the other? This guarantees that h (whatever it is) must increase away from the boundary at which the condition is applied.
The flow vanishes at the boundary because the slope of isopycnal is zero. As a result, we consider this assumption at the boundary. Thanks to using numerical model results, the velocity field also confirms this theory when the slope of isopycnal is zero, the velocity is also zero. Hence, it is a guarantee for us to check our analytical method results.

Line 15 - needs to explain that 1/beta is considered the relevant Rossby radius for this problem.

Corrected.
Page 19, lines 36-37 - you state that vorticity is predicted correctly and that the model confirms a prediction, but present no evidence for this.

This was discussed above.

Page 20 bottom line - I think you are also assuming that |L1|=|L2|.

Yes. However, we defined new parameter L in the obtained new formula, which means that this definition $L = \frac{|L_1|+|L_2|}{2}$ is enough and not contrasted with|L1|=|L2|.

Page 21 - why assume the outflow from the middle basin to be the relevant flow to calculate the flushing time? Surely the inflow is more relevant. What is the meaning of h in Figure 18?
That's a good question. It depends on how observer look the phenomena when we look the flow in the middle basin is outflow (because getting out from middle basin) and you consider southern basin which indicates that the flow is inflow. It shows that you and we describe one phenomenon in a different view.
About *h*, in the new version, we use h' because it is different from h. h' is the distance of isopycnal from bottom in $L_1$ and $L_2$.
It should also be mentioned that the densification due to evaporation and freezing mainly occur in the northern shallow basin attached to the middle basin, hence the deep overflow from the middle basin to the southern one is often usually used as in case of semi-enclosed sea as Mediterranean sea.

Figure 19 - contour labels are virtually unreadable. The contour interval is not anything obvious, and the critical contour used for calculations (1008.78) should be highlighted (it is not even one of the plotted contours). In fact, this contour doesn't reach the bottom anywhere in the January plot, making it impossible to identify meaningful parameters

in this case.

About the choice of 1008.78 kg/m$^3$, this contour has been selected with a detailed study for all months. The major problem is the position of the contour which changes for different months. If we consider the contour 1008.9, it is good for some months like January, but it is not useful for November because the maximum contour is 1008.8 for this month. I agree with you about January, we can choose different counters for each month to solve this problem. However, this method cannot be useful to compare all month in terms of $Q_v$. For this reason, we try to use the same method for calculation, although we accept your opinion. This create an error in limited months such as January. For this reason, we show how the error can occurs using this assumption (Fig.6).

[Figure]

Fig. 6: The black arrows shows that the points which is used to calculate $Q_v$. Near the bottom is ok without any errors. However, 1008.78 dose not reach the bottom but we assume that it reachs the bottom.

Table 5 - I can't find a way to get values from Fig. 19 that agree with 2L values in this table. Are these actually L?

The first step is the calculation of $L_1$ and $L_2$. Then L=L1+L2/2.To examine the accuracy of the calculation of L, the length of the strait is between 10 to 59 km in deep parts. Based on table 4, 2L is 19 to 34 km. It is rational because 2L is smaller than 10-59 km. 2L=19 km is for November because the contour 1008.76 is found more suitable for calculation of L.

---

## Author Comment (AC2) · 25 Jun 2018

Many thanks to anonymous Referee for comments on "Some aspects of the deep abyssal overflow between the middle and southern basins of the Caspian Sea" by JavadBabagoliMatikolaei et al

Our answers and further clarifications are as follows:

Page 2 Lines 2-18 are not made to relate to the Caspian. I think an alteration might be made at line 13, e.g. ".but also in ventilation of semi-closed and closed basins, e.g. the Caspian Sea. Study . ." Lines 17-18 seem misplaced; they are not made to relate to the previous or following text.

Due to the fact that in the Caspian Sea, little research has been done on this topic, we have listed other areas of research for this section.
Regarding the study of Bidokhti and Ezam (2009), though it is not related to the Caspian Sea, we have used the results of this article in section 3-2 in our paper. As a result, we introduce this paper in the introduction.
Coming to an alternation in this regard, your comment has been implemented.

Page 3 Line 16. "7-10" does not quite correspond with figure 1. Line 19. "16" does not correspond with lines 15-16 or figure 1.

Ibrayev et al., 2010 showed that the mean values for temperature in 3 basins for many years. However, we plot the temperature of each basin with hourly data collected by Modis sensor. As a result, the results of both studies should not be the same. Due to your comment, we add some more explanation for the reader to understand this difference.

Figure 2. Both sides would benefit from a distinct coastline. The left side should have the same latitude and longitude scales.
Corrected

Page 6. Line 3. "layers" not "levels". Line 14. I am not convinced by "are rather consistent with observations". However, currents here are not very relevant to the sill and overflow there. Line 16 "as can be expected" and line 17 "interpolation". The uncertainties due to the model grid might be estimated by comparing the variance of (model-observation) with the variance of (difference between adjacent model grid points). Figures 7 and 8 are probably better evidence that the model is working OK for the purpose of this study.
Two points should be taken into consideration. The first point is the model predicts the flow behavior well. The second point is the value of the velocity which is not exactly the same as that of observation data. This fact is quite natural because the model has assumptions and does not model some phenomena exactly. If a numerical model is exactly the same as the observations, then surely the performance of the model should be skeptical. According to what was mentioned, measurement data is very limited and scarce in this water basin and the velocity data is one of the best measurement data in the Caspian Sea, as this data have already been used in published paper in OS. As there are not similar ADCP data near the Absheron sill we used a numerical model in

this study. The purpose of this section is to validate the model so that we can evaluate the accuracy of the model and used its results in the analytical models.

Page 7. Lines 4-5. I think this sentence "This.Absheron." does not add information. Lines 5-6. I think "From . . derived." belongs at the beginning of section 3. Line 7. Refer to figure 9 as well
Corrected

Page 8 Lines 37-41. I think this belongs in section 3 before section 3.1.
Corrected

Page 10. Line 5. "No pressure gradient". There are horizontal gradients of density implying horizontal gradients of pressure. However, I think equations (1) can be OK if understood as in coordinates parallel to the slope. Equations (1). I think there might be some comparison with the model of Shapiro and Hill (1997) J Physical Oceanography, 27(11), 2381. It is very similar albeit steady-state. Line 19. In section 3.1 "re" and "rb" only appear summed as "rb+re" and a symbol for "rb+re" would be useful.
When it comes to the model of Shapiro and Hill (1997), there is some common ground between the two papers (and this paper). However, Shapiro and Hill use stead-state assumption to solve the momentum equation. This paper solves a momentum equation for upper and lower layer because they consider the diffusion term to link the upper and lower layer dynamic. This work is very similar to Cenedese, C. J., and Whitehead, A., 2002. Cenedese, C. J., and Whitehead presented an analytical model based on two layers under steady condition with stirring diffusion.

*(Cenedese, C. J.,and Whitehead, A., 2002. "A dense current flowing down a sloping bottom in a rotating fluid", J. Phys. Oceanogr, Vol. 34, 188-203).*

 Apart from this, Hughes and Griffiths presented a simple convective model with effects of entrainment. They investigated roles of vertical mixing and surface buoyancy fluxes in the dynamics of the global overturning circulation . Please see: *Hughes, G. O., & Griffiths, R. W. (2006). A simple convective model of the global overturning circulation, including effects of entrainment into sinking regions. Ocean Modelling, 12(1-2), 46-79.*
Based on previous work, it means that you can solve the momentum equation analytically with some assumptions.
As a result, we should choose either steady with stirring diffusion or non-steady without diffusion to solve momentum equations, as the path of the flow was important for us, the second method was used in present work.
About different symbol for $r_b$+re, this's good point, however, we tried to see separately the effect of friction and entrainment because $r_b$ and $r_e$ do not have the same value. For this reason, we defined two parameters to show the value of each quantity separately.

Page 13. Table 1 columns could be fitted to contents so that row 4 is all on one line. Line 14. "direction" not "horizontal". Line 18. Word order better ". .flow is trapped after about 10 km. . ."
Corrected

Figure 12. The x and y scales in the left panel differ. Ideally they should be the same but if not the caption should say that they differ.

Because the two-axis scale is not the same. If their scale is identical, part of the flow path will not be shown. Due to your comment, some explanation has added the figure 12.

Page 15. Lines 4-10 should somewhere state the assumption of steady flow. Equations (5), (6). In the integral exponents (of e) I think the integrand should be r/u where u has anoverbar. "r" needs definition. Figure 15. In the left panel, left axis, the variable should be ζ Page 16 line 39 to page.
Corrected. The formula extract from Falcini and Salusti (2015).

17 line 1. ". .the graph shows decreases from I to IV . ."
Corrected

Page 17 Line 5. Omit first "sea". Line 6. "similar assumptions" – but also steady which should be stated. Line 8. "No mixing could exist" - not true; no mixing is an assumption
Corrected.

Line 10. "0.00002 s**-1" should be related to 0.003|U|/H,and values of U, H.
Corrected. The value of each parameters from which we estimated $r_b$ are discussed in section 3.1.1.

Page 18 Line 4. "short distance between D and V" should be shown on the same figure. How are locations I, . . V defined? Lines 5-6. "D is the location at which the current is trapped by the topography". What about C and E for example? Line 15. To obtain R from (11) involves a value for potential vorticity Π. How is this estimated?
In the new version of the paper, D and V are shown in the figure 1 to compare them with each other. The most important point in choosing these points was the distance between the two points, which is about 30-50 km. Between I and III due to the variations in the depth of the Strait, it was considered to be about 30 km, and the vorticity in these points changes more. Between III and V, depth changes are less, and therefore stretching term in vorticity (Eq. 6) equations have less effect. As a result, for this point, the distance between points is considered 50 km. About transect E, we used the Rossby radius of defamation changes rather than transect C (see page 19, line 33-36 in main paper). Transect C is the point on the sill and can be useful to understand how much the water sinks when comparing with transect A. We discussed that in the previous section in the present text.
We assume that vorticity is the same in V~D and to estimate vorticity we use figure 15. Π is estimated based on Eqs. 5 and 6 which is extracted from Falcini and Salusti(2015).

Page 19 line 36. "The numerical model . .confirms this prediction." The reader cannot infer Π quantitatively from figure 9.
Yes. This sentence is omitted and the accuracy of the analytical and numerical model is quantitatively compared in table 3.

Page 20. Equation (13). This form for h differs from (10). Equation (14). ")" missing after exponentials. Last line. "If we assume . ." does not give (15) directly: L2≠ L1Need to say ". .we approximate (14) by". Then it makes sense to compare the values given for Qv by (14) and (15) (page 21 line 5)

Your comment is quite correct and has been modified in the new revision. However, we considered a new definition for isopycnal in this part because the Strait is narrow and the shape of isopycnal lines in the sill is simpler than its shape in the southern Caspian Sea. For this reason, the terms in the h' are simpler than h also inferred than the boundary condition which is difficult to calculate the non-changing value in h profile.

Page 21 lines 10-11. Better to move "are calculated" to before "below"

Corrected.

---

## Author Response (AR2)

Dear professor John M. Huthnance

Many thanks for the points on "Some aspects of the deep abyssal overflow between the middle and southern basins of the Caspian Sea" by Javad Babagoli Matikolaei et al. Some points are very useful for us to improve our study. We try to answer and address one by one each comments as follows:

Dear Javad Babagoli

Thank-you for sending me this revised version. I think there are still some improvements that you should make before it is reviewed again.

The main point among the details below is that I think there needs to be more explanation for the two analytic models: How are they more useful than the numerical model? What are the differences in physics between the two analytic models? How good are the assumptions enabling analytic treatment?

Yes. Based on your opinion, we add a new section in page 21, section 3.3.

Detailed comments and corrections.

Page 2.

Lines 13-14. Better ". . He pointed out such flows around world that may lead to . ."

Corrected.

Page 2, lines 13-14.

Line 30. Add "," after "tracers".

Corrected.

Page 2, line 29.

Line 31. Better ". . year by year, dominated . ."

Corrected.

Page 2, line 30.

Page 3.

Line 22. Better ". . difference for these two points . ."

Corrected.

Page 3, line 24.

Line 24.  Delete "is" at start.

Corrected.

Page 3, line 26.

Page 4 line 9.  Better ". . 2005 to validate the Numerical"

Corrected.

Page 4, line 9.

Page 5.

Line 10.  "Compression" -> "Comparison"

Corrected.

Page 4, line 13.

Line 12.  Better "spacing of CTD stations, distance is plotted in kilometres on the top . ."

Corrected.

Page 5, line 11.

Lines 13-14.  "mainly due to the low resolution of observational data".  But there is more detail in Figure 4(a) than in 4(b).  Are these figures perhaps (a) model and (b) observational (opposite to caption statement)?

4a and 4b are the observation data and numerical simulation respectively. In the new version, at the end of each sentences, we emphasize which one is related to numerical and simulation.

Page 5, line 13-14.

Last 2 lines.  "distance between two" (twice).

Corrected.

Page 5, line 17-18.

Page 6.

Line 6.  Better "To achieve our goal, we ran the numerical model and used these data."

Corrected.

Page 6, line 3.

Line 13.  "coastlines and bathymetry data with 0.5º ×0.5º resolution are acquired from GEBCO".  Please check this stated resolution.  Figures 6, 9, suggest finer resolution.

Corrected. Thank you so much. 0.5′ (30 seconds) is right rather than 0.5°.

Page 6, line 11.

Lines 26-27.  Better ". . validation and are not related to deep flow near the Absheron Strait. For the latter, ADCP data in the Strait would be very useful . ."

Corrected.

Page 6, line 25-26.

Lines 29-30.  This last sentence of the paragraph is very unclear.  Maybe "This similarity relates to timing of flow increase and decrease rather velocity magnitude."?

Ok. Corrected.

Page 6, line 28.

Line 1.  Better ". . data at the location of"

Corrected.

Page 6, line 31.

Lines 23-24.  Better ". . Strait is correct as is reflected . ."

Corrected.

Page 7, line 21-22.

Page 9.

Line 22.  ". . closed . ."

Corrected.

Page 9, line 22.

Lines 22-23.  Better ". . Caspian Sea.  However, accurate information about the connection is not accessible for whether to include the higher-salinity source in the numerical model . ."

Corrected.

page 9, line 22-24.

Line 27.  Better ". . 200-300 m in the middle basin in the numerical simulations, . ."

Corrected.

Page 9, line 27-28.

Line 33.  "isopycnals".    ". . (see 3a, b and 4a).  Some other"

Corrected.

Page 9, line 33-34.

Lines 34-35.  Better ". . Caspian sea (e.g. Terziev et al. (1992) and Ibrayev et al. (2010))."

Corrected.

Page 9, line 35.

Page 10 line 35.  Delete "but".

Corrected.

Page 11, line 1.

Page 11 lines 1-3.  The sentence "One of . . Sea." looks like an aim of the paper which should be in the Introduction.

Corrected. We moved the sentences to the introduction.

Page 3, lines 11-12.

Page 16.

Lines 20-21.  Better ". . solved analytically using sufficient assumptions related to the research aims.  In comparison with this work . ."?

Corrected.

Page 16, lines 20-21.

Line 24.  "profiles" is unclear, usually referring to depth dependence but here there are 2 layers at most.

Ok. Now, we use velocity section instead of profiles. This was exactly used by Cenedese et al (reference: page 191: Figure 2)

Page 16, line 24.

Line 26.  "and also sinking process are more important when the overflow moves over the Strait." is not clear.  "more" implies a comparison; between what and what?

Corrected.

Page 16, lines 25-26.

Line 30.  "ventilated"

Corrected.

Page 16, lines 30.

Line 31.  "can sink up to 600-700 m".  Is there a limit to the sinking in this model?

No, it is not a limitation for the model and the 400-500 is correct. However 400-450 m is more appropriate because the depth of sill is about 200m and the sinking is about 200-250 m. as a result the water can sink to 400-450 m in this process.

Page 16, lines 30-31.

Line 32. Better ". . Black sea which lacks such ventilation and is almost non-productive . ."

Corrected.

Page 16, lines 33.

Page 17. There still needs to be motivation for TWO analytic models.

Corrected.

Page 17, lines 5-13.

Line 4. Define PV

Corrected.

Page 17, lines 12.

Lines 11-13. How do the assumptions here compare with those in section 3.1?

Corrected.

Page 21, section 3.3.

Line 28. "mean". How are the vorticity and PV averaged (in space and time)?

The equation 5 is as the Eq.13b in Falcini and Salusti (2015). In the new version, we add some note to clarify the reasons of using averages in the Eqs. (See page 21, section 3.3). Here we try to explain more about this. Based on the Falcini and Salusti paper, they obtained the vorticity equation as follows:

$$\frac{\zeta}{f} = e^{-\int_0^{\xi} \frac{X}{u} dx} \left\{ \frac{\zeta_0}{f} - \int_0^{\xi} e^{\int_0^x \frac{X}{u} dx'} \frac{1}{u} (\text{div}\, u)\, dx \right\}. \qquad \text{(a)}$$

To obtain this formula, they use many assumption. After that due difficulty of calculation $h(\xi, \psi)$ and u(x,y) from data, they used some assumption and prove the following equation as cross sectional average.

(b)

$$\frac{\overline{\zeta}}{f} = e^{-\int_0^\xi \frac{X}{\overline{u}}dx}\left\{ \frac{\overline{\zeta}_0}{f} - \int_0^\xi e^{\int_0^x \frac{X}{\overline{u}}dx'}\frac{1}{\overline{u}}(\overline{\mathrm{div}u})\,dx \right\}$$

After some mathematical operation the Eq.5 is obtained. About how the Eq. b is obtained from Eq. a please see page 402, **Appendix A** in Falcini and Salusti(2015).

Line 32. Δh and Δx are not defined and not in (5), (6).

Corrected.

Page 18. Lines 6-7.

Lines 9-10. Better ". . although in January the vorticity is a minimum among other months . ."

Corrected.

Page 18. Lines 22-23.

Lines 11-12. "changes of Π over the sill (from I to III) are about . ."?

Corrected.

Page 18. Lines 24-25.

Line 15. Better ". . Fig. 15 shows an important point that the changes of"

Corrected.

Page 18. Lines 28.

Line 19. "pressure gradient" (reverse order of the two words)

Corrected.

Page 19. Line 3.

Lines 21-24.  What is the basis of these statements about the width of the flow?  It does not appear in the equations (5) or (6).  If these statements are based on the numerical model, please say so.

In new version, we explain our method in page 17, lines 12-13. Based on your comment, we emphasize.

 Page 19, lines 6-7.

Line 24.  "The results" – of what?

 Corrected.

 Page 19, line 8-9.

Page 19.

Line 8.  Not "figure 16" which is transect V not near the Cape.

Ok. Corrected.

See page 19, line 18.

Line 9.  Insert "(figure 16)" after "May"?  But neither figure 16 or 17 shows PV directly.

Corrected. We don't want to show the PV in these figures. In general, we discuss the behavior of the flow near the cape.

We use the potential vorticity in the upstream because Stern shows that his method is based on the upstream parameters of the flow. For this reason we calculated the vorticity and potential vorticity from I to V which is located before the cape. We explain it in page 19, lines 22-24.

Line 21.  Better "separation.  A more rigorous criterion is needed . ."

 Corrected.

 Page 19, lines 30-31.

Page 21 lines 8, 11 and page 22 lines 8, 10.  I think you mean Figure 18 each time (page 23 line 23 "Fig. 18" correspondingly).

 Corrected.

 Page 22, lines 18,21…

Page 22.

Line 5.  "with less assumptions."  What is/are the extra assumption(s) in (10)?

Corrected.

Page 23, line 10.

Lines 15-16.  Better ". . table 4, the numerical model value is greater than that of the analytical model. This underestimation by Eq. 10 can be due to the fact . ."

Corrected.

Page 23, lines 20-21.

Page 23.  Caption for figure 18 needs to explain 18c.

Corrected.

Page 24, lines 27-29.

Page 24.

Line 12.  Better ". . middle basin, like deep convection in"

Corrected.

Page 25, lines 15.

Lines 21-22.  Better ". . The Rossby scale width of the flow varies for different seasons.  . ."?

Corrected.

Page 25, line 25.

Line 24.  Delete "that are used".

Corrected.

Page 25, line27.

---

## Author Response (AR3)

Dear professor John M. Huthnance

Many thanks for the points on "Some aspects of the deep abyssal overflow between the middle and southern basins of the Caspian Sea" by JavadBabagoliMatikolaei et al. Some of these points are very useful to improve the paper. We try to answer and address one by one of each comment as follows:

I think there is still some lack of clarity about the purpose of the whole paper, and certainly about the role of the two analytic models. Fair comment is made that with paucity of observational data, models are wanted to increase the information/evidence available. Given the dependence of the analytical models on some input from the numerical model, what is their added value? There are 2 or more possibilities.
1) Analytical models may need less observational data as input than would a 3-D numerical model. I don't think this is stated although there is mention that the methods may help where there is paucity of observations.
2) Comparison with the numerical model might indicate that the simplifying assumptions in an analytic model are valid. This could lead to an understanding of the most important dynamics involved. Such an advance is mentioned in passing but there is no demonstration of a good match between numerical and analytical model results, which in any case might be "forced" by the initial input of numerical model information to the analytical model.

Based on these comments, we conclude that it should be better to omit the first Analytical model and add many extra sentences to clarify the main purpose of this paper. Then we put emphases on the numerical results.

The following detailed comments include many instances where the intended meaning or logic is not made clear.
Line 18. For what reason?

Corrected. See page1, line 18.
Line 24. Does "this work" refer to Falcini and Salusti (2015) or the present manuscript?
Corrected. See page 1, line 21.
Page 2.
Line 3. "thermocline" should be "thermohaline"?

See page 2, line 3.
Lines 3-5. I think this sentence could be omitted given the following sentence.

Omitted.
Line 13. "near shelves" should be "near-shelf"
See, page 3, line 12.
Page 5.
Line 16. Better ". .very coarse for showing the overflow. . ."

Line 19. "high regulation" should be "fine-resolution".

Page 6 lines 23-24. This sentence lacks some logic. The purpose of running the model is the lack of observational data. The ADCP data are not for the last year but only for three months according to line 21. The sentence seems to imply that the purpose of running the model is to check the accuracy of its performance, but that is not the main reason in the rest of the manuscript.

The sentences are changed. See page 6 lines 22-25

Page 7 lines 21-22. This sentence is very unclear. Do you mean that the simulations and the observations both confirm the existence of a deep flow in Absheron Strait? (Figure 4.)?

The sentences are rewritten. Page 7, lines 21-22.

Page 9.

Lines 11-14. This should have a reference to literature, or is it a result of the numerical model? Please clarify.

Now it is clear. See page 9, line 10.

Lines 31-32. This looks much like lines 12-13.

Lines 31-32 are omitted.

Top line. Better ". .of the overflow is about . ."

Corrected.

Figure 8 caption, last line. "similar" as regards depth, but not the slope.

Corrected.

Page 12.

Lines 5-6 and 11-13. These are at present divided by equations (4) but should be brought together and put in logical order. Please state the line 12 "input parameters" – g', up, vp but not the others in Table 2?

The formulas are omitted in this version.

Line 14. Figure 11 is transect II further downstream. How does this contribute to the input parameters?

This section has been changed and now both transects are considered simultaneously in Fig.11.

Page 13 figure 10.

Page 15 lines 16-17. Unclear logic; why should the method (what method) depend on N-S direction of flow?

The page is omitted.

Line 23. I do not understand "stirring diffusion". Maybe "stirring and diffusion" or just one of the two.

Lines 27-28. ". . In this paper, it is shown that the dense water is firstly formed . .". No! it is stated several times but not shown in this manuscript.

Lines 33-34. This sentence seems out of place. There are a few sentences like this that might best be brought together in an "impacts" sub-section.

The page is omitted.

Line 5. I do not understand "changes in flow in its perpendicular direction". The three directions are vertical, along-flow or across-flow (= lateral).

This is about the previous model and now it is omitted.

Line 6. Not "because the ventilation could be addressed". Maybe "for their influence on ventilation"?

Lines 12-13. "we need to use an analytical model" is not convincing when there are numerical model results.

In new version, we add some notes to clear this in page 14.

It should be noted that in previous version, we used three analytical models. Your comment is completely right about using analytical model when there are numerical model results. However, in the first model, we investigate the path and the sinking process. Both of this phenomena can be studied by numerical model without using extra formulas. For this, we accept your comment. But the second formula is vital to calculate vorticity and potential vorticity. It means that the model outputs give velocity components (u,v,w) and temperature, salinity and density. Using the numerical outputs and a formula we can estimate the vorticity and potential vorticity. In other words, the numerical model directly does not calculate the vorticity.

Line 19. "They" – is this Falcini and Salusti? Needs to be stated because the present implication is Whitehead et al.

please see page 16, line 15.

Page 18.

Line 3. "receptively" should be "respectively"

Corrected. page 17, line 6.

Lines 22, 23. The symbol for PV should be in italics.

Corrected. Page 17, lines 26-28.

Line 29. "because the depth of water is changing . . (stretching term)" applies to vorticity but NOT to potential vorticity.

Corrected. Page 18, line 2.

Page 19.

There should be a reference somewhere to figure 16.

Please see Page 18, line 18.

Lines 18, 19, 20. "Nov." should be "November"

Corrected.

Line 29. Not "over the cape". Maybe "north of the cape"?

Corrected. Page 18, line 34.

Page 21.

Line 13. Delete "an increase in"

Line 17. "which does not account for mixing in the momentum equation". But you have E*.

Lines 18-19. This sentence is very unclear. Perhaps this is my point 2 near the top of my comments.

Line 20. "it" -> "vorticity". However, this does not explain why the analytic model rather than the numerical model is used. There is a statement about this on line 22 but it is very weak. From where do you get the value of ζo?

Lines 25 to 30 are confusing. You mix discussion of the first model (lines 25, 27) with discussion of the second (lines 26-27, 28-30). You say (no) "approximation such as balancing between forces" but (a) any momentum equation is a balance between forces, (b) your first model has zero pressure gradient, (c) the Falcini and Salusti (2015) model neglects time-dependence. Also lines 28-30 need to be clearer that you are discussing the model that you used. Line 30 "in our method" needs to be clear that it is the second model.

Line 31. "on their assumption" – what assumption.

Line 35. ζ not ξ

Line 36. ". . between the two . ."

This page is omitted. However, ζo is added on page17, lines 5-6.

Page 22.
Line 5. Better "southern basin entrainment is negligible (due . ."

This is omitted.
Line 22. I think you substituted equations (8) as integration limits in equation (7). What "assumptions"? see page 20, line 30.

Page 23.
Lines 3-5 "If we assume . .we have . . (10)" No! (10) is an approximation assuming L1 = L2. Lines 8-10 do not explain this properly. I think the difference between (10) and the accurate value of Qv depends on α and L2/L1 (at least, possibly other variables as well).

Lines 26-27. This sentence is unclear. If there were a special contour for each month, why would Qv be calculated "under the same conditions for all months"?

Corrected. Page 22, lines 9-10
Line 27 to top of page 24. If the flushing time is estimated based on the . . numerical simulations, what is the purpose of equations (8) to (10)? Perhaps you mean ". .is also estimated . .". Which method is used for the values in Table 4?

By the outputs of model which incude the components of velocities and temperature, salinity and density, we have to use a formula like this

$$Q_V = \int v ds$$

The model give us v, but ds is the main problem for us. As a result of this we have to calculate ds by using some assumption. However, in equation 6, again we have to calculate ds with some assumption. The differences between two methods is v, in equation 6 which is considered the same.

Page 25.
Line 9. I think this section might be "Conclusions and consequences" (or some similar extra word) because you discuss impacts which are not really conclusions of your study (page 26 lines 3 - 9).
Lines 13-17. These are probably true but not really conclusions of your study.

Corrected.

[revised manuscript text omitted]

---

## Author Response (AR4)

Dear professor John M. Huthnance
Many thanks for the points on "Some aspects of the deep abyssal overflow between the middle and southern basins of the Caspian Sea" by Javad BabagoliMatikolaei et al. Some points are very useful for us to improve our study. We try to answer and address one by one each comments as follows:

Many places.  Please check spelling: Absheron or Apsheron?
Corrected. (Absheron)
Line 13.  ". . as there has been only a little research . ."
Corrected
Line 16.  ". . investigating the dynamics . ."
Corrected

Lines 16-17.  Delete either "here" or "in this section".
Corrected.
Page 9, figure 7 caption first line.  Better ". . between the north-south cross-sections of mean temperature"
Corrected.

Page 10.
Line 17.  ". . comparisons. We then use . ."
Corrected.

Line 18.  Delete "Although" or "but"
Corrected.
Line 21.  Better ". . there is only a short distance . ."?
Corrected.
Line 22.  Delete "those".  ". . transects such as IV."
Corrected.
Line 23.  ". . section, we investigate . ."
Corrected.
Lines 24-25.  Better ". . 105 m for January and 125 m for September.  These depths are the mean of the maximum and minimum . ."?
Corrected.
Line 30.  "flashing" -> "flushing"
Corrected.
Page 11.
Line 10.  "closed" -> "close"
Corrected.
Line 11.  Can omit "Major other"
Corrected.
Line 13.  Delete "were"
Corrected.
Line 16.  A "latitude line" means east-west; do you mean this?  Maybe delete "along a latitude line" since the reader knows this is water moving from transect I to transect II.
Corrected. Latitude line is omitted.
Line 16.  ". . for this sinking depth variation . ."
Corrected.
Page 13.

Line 5.  ". . section, we investigate . ."
Corrected.
Line 11.  "parameters" -> "variables"
Corrected.

Lines 13-14.  Delete "although" or "however".
Corrected.

Line 18.  "the Cape depending on the outflow properties; its buoyancy varies . ."?
Corrected.

Lines 19-20.  Better rearranged.  "behavior the vorticity and potential vorticity of the flow column upstream of the Cape is linked to the separation of the flow from the Cape, as previous works (Ezam et al., 2012; Stern, 1980) have shown.  Here . ."
Corrected.

Lines 31-32.  Please do not use u and U for the same quantity.  If U is the magnitude you do not need "|..|" in the formula on line 31.
Corrected.

Page 15 lines 13-15.  This is very unclear.  How many layers are there.  In how many layers is there flow?  Please state clearly what layers there are, in which layer(s) is there motion, and what you mean by the "effect of the upper layer".  Lines 20-24, (1) and (2) have no mention of different layers and rather suggest only one deepest moving layer, possibly with a layer above with no flow.
In their work, firstly they used three layers for Momentum and mass formulas to achieve some equations for friction and entrainments; then they concentrate in third layer (dense flow). In our paper, we used the final result of their paper 1 and 2.
I tried to rewrite this section.
Lines 8-9.  This sentence is not needed.  But please do make sure that all symbols are consistent in this manuscript!
Corrected.

Line 14.  "stream tube" (spelling)
Corrected.
Line 28.  Π in italics please.
Corrected.

Line 23.  Delete "rather"
Corrected.

Lines 27-28.  Please remove gap so formula appears on one line.  Delete ")" after "f".
Corrected.

Page 18 figure 4 caption.  Can omit "which is rather close to the bottom"; you explain k at the end.
Corrected.

Line 5.  Can omit "will be"
Corrected.

Lines 15-17  Better ". . results. Although use of observational data is common for deep flow volume flux estimation, we do not have ADCP data across the Absheron sill. . ."
Corrected.

Line 18.  ". . across Absheron Strait."
Corrected.

Line 27.  ". . and h', (deep . ."  [order of symbols]
Corrected.

Page 20.
After (6) you must say that (6) is an approximation for $|L1| = |L2|$.  Otherwise (5) and (6) would be the same and lines 14-15 are meaningless.  $2 \, Sinh(\alpha L) \neq (e^{\alpha L1} - e^{-\alpha L2})$, the term in b is missing from (6) etc.
Corrected.

Line 16. Better ". . geostrophic balance between v and h' is assumed as Ro ~ 0.1 . ."
Corrected.

Line 21.  ". . -150 m which is the approximate depth . ." [But earlier you said 180 m sill depth].
Corrected. Typing error.

Page 21 line 12.  "on" -> "in".
Corrected.

Page 23.
Line 10.   ". . section 4.1, we estimate . ."
Corrected.
Lines 15-16.  "This is the main reason" – you have not given a reason!  I think you mean "The main reason . . is that this deep flow carries"
Corrected.

Lines 19-20.  This sentence adds nothing – omit.
Corrected.

Line 33.  Omit "although"
Corrected.

Page 24.
Line 6.  "due to the fact that" -> "and"?  Present text implies that the eddies cause the separation but section 3.2 implied that upstream potential vorticity was a control.
Corrected. Our ambition was upstream potential vorticity.

Line 10.  ". . present and that pollution can spread . ."
Corrected.

Line 15.  ". . basin it finally overflows . ."
Corrected.

Page 25.
Line 7.  Delete "that"

Corrected.

Line 9.  Delete "process"
Corrected.

[revised manuscript text omitted]

---

## Author Response (AR5)

Dear professor John M. Huthnance

Many thanks for the points on "Some aspects of the deep abyssal overflow between the middle and southern basins of the Caspian Sea" by Javad BabagoliMatikolaei et al. Some points are very useful for us to improve our study. We try to answer and address one by one each comments as follows:

Technical corrections.
Page 2 line 10. Delete "the" at end. [It tends to suggest that all shelves have along-slope currents].
Corrected.
Line 7. Better ". . Caspian Sea that has only been touched on in the"?

Corrected.
Line 11. Better ". . overflow. This paper is divided . ."

Corrected.
Line 16. "investigating" -> "of".
Corrected.
Page 7 line 23. Omit "At the beginning . . pointed out that"
Corrected.
Page 8.
Lines 14-15. Mention of Denmark Strait seems a bit odd here but you refer to it again later. If you want to keep ", similar . . of the DS" then I think you should give a reference here.

Corrected.
Line 17 near end. "on" -> "in"

Corrected.
Line 22. Omit "Over the years"
Corrected.
Page 10 line 19. ". . isopycnal is . ." (space between words)
Corrected.
Page 13.
Line 4. ". . (2003); they used . ."

Corrected.
Line 8. ". . affecting the flow dynamically."

Corrected.
Line 12. ". . can be shown by numerical model simulation."

Corrected.
Line 18. ". . behaviour of the vorticity"

Corrected.
Page 14 line 7. Add space between superscript "-1" and "based".
Corrected.
Page 15 line 15. ". . concentrate on the third layer . ."

Corrected.
Page 16 line 11. "1). Thus rb = . . ); re = . . (transect III) based on"
Corrected.
Page 17.
Line 18. ". . from the cape . ."

Corrected.
Line 20 end. ". . spring (May; Fig. 14)."

Corrected.
Line 31. "in" -> "of" ; "," after RD should be full size.

Corrected.
Line 32. ". . respectively). Fig. 14 indicates separation . ."

Corrected.
Line 33 & Page 18 line 1. ". . cape, more pronounced in November. Considering . . cape, the Stern criterion . ."
Corrected.
Page 18 line 10. "Absheron" (spelling).
Corrected.
Page 19.
Line 4. ". . basins' . ."

Corrected.
Line 6. ". . discussed . ."

Corrected.
Line 17. Delete 2nd "Absheron"

Corrected.
Line 21. ". . its geometry and the upper surface of the dense overflow . ."

Corrected.
Line 26. Delete "," before "(" and after ")"
Corrected.
Page 20.
Line 13. "1", "2" should be subscript.

Corrected.
Line 14. Delete "as"

Corrected.
Line 17. "section"

Corrected.
Line 22. ". . which is the approximate . ."

Corrected.
Line 23. ". . basin). Then if . ."

Corrected.
Page 21 lines 12-13. ". . in oceanic contexts without ADCP data, for example the Persian Gulf . . CTD data in the Hormuz"
Corrected.
Page 23.
Line 10. ". . Caspian Sea basins' flushing times because they are very important . ."

Corrected.
Line 11. Delete "as"

Corrected.
Line 16. ". . that oxygen is carried from . ."

Corrected.
Line 25. ". . sea bed in Absheron . ."

Corrected.
Line 32. "activities at the bottom . ."
Corrected.
Page 24.
Line 1. ". . oil drill holes on Absheron . ."

Corrected.
Line 2. "through" -> "over"

Corrected.
Line 4. Add space before "("

Corrected.
Line 5. "substantial" -> "significant"? Delete "and climate"? "of" -> "or"

Corrected.
Line 6. Delete "rates"

Corrected.
Lines 6-7. ". . Based on the path of the deep flow and the eddy formation near SefidRud Cape, it can be concluded . ."

Corrected.
Line 10. ". . like sea level . ."

Corrected.
Line 11. Better ". . raised regarding the effect . ."

Corrected.

Line 12. "Based on section 2.1, it is mentioned that the" -> "At present"?

Corrected.

Line 13. Delete "it is accepted that"

Corrected.

Line 15. ". . volume of water in the deep overflow . ."

Corrected.

Line 16. "words, warming probably . ."

Corrected.

Line 22 end. ". . (b) The map" (add space after ")").
Corrected.

Lines 15-16. Better ". . vorticity of the trapped current over Absheron sill."

Corrected.
Line 17. ". . m3 of water per year in this abyssal flow . ." (add space after superscript 3)

Corrected.
Line 18. ". . which is of the same order . ."

Corrected.